# DON'T TRUST YOUR EYES: ON THE (UN)RELIABILITY OF FEATURE VISUALIZATIONS

## ABSTRACT

How do neural networks extract patterns from pixels? Feature visualizations attempt to answer this important question by visualizing highly activating patterns through optimization. Today, visualization methods form the foundation of our knowledge about the internal workings of neural networks, as a type of mechanistic interpretability. Here we ask: How reliable are feature visualizations? We start our investigation by developing network circuits that trick feature visualizations into showing arbitrary patterns that are completely disconnected from normal network behavior on natural input. We then provide evidence for a similar phenomenon occurring in standard, unmanipulated networks: feature visualizations are processed very differently from standard input, casting doubt on their ability to "explain" how neural networks process natural images. This can be used as a sanity check for feature visualizations. We underpin our empirical findings by theory proving that the set of functions that can be reliably understood by feature visualization is extremely small and does not include general black-box neural networks. Therefore, a promising way forward could be the development of networks that enforce certain structures in order to ensure more reliable feature visualizations.

## 1 INTRODUCTION

A recent open letter called for a "pause on giant AI experiments" in order to gain time to make "state-of-the-art systems more accurate, safe, interpretable, transparent, robust, aligned, trustworthy, and loyal" (Future of Life Institute, 2023). While the call sparked controversial debate, there is general consensus in the field that given the real-world impact of AI, developing systems that fulfill those qualities is no longer just a "nice to have" criterion. In particular, we need *"reliable" interpretability methods* to better understand models that are often described as black-boxes. The development of interpretability methods has followed a pattern similar to Hegelian dialectic: a method is introduced (*thesis*), often followed by a paper pointing out severe limitations or failure modes (*antithesis*), until eventually this conflict is resolved through the development of an improved method (*synthesis*), which frequently forms the starting point of a new cycle. An example of this cycle are saliency maps: Developed to highlight which image region influences a model's decision (e.g., Springenberg et al., 2014; Sundararajan et al., 2017), many existing saliency methods were shown to fail simple sanity checks (Adebayo et al., 2018; Nie et al., 2018), which then spurred the ongoing development of methods that aim to be more reliable (e.g., Gupta & Arora, 2019; Rao et al., 2022).

In contrast to saliency maps and attribution methods like GradCAM (Selvaraju et al., 2017), LIME (Ribeiro et al., 2016) and SHAP (Lundberg & Lee, 2017) where the field has developed a relatively good understanding of their reliability, another central mechanistic interpretability method currently lacks good sanity checks: feature visualizations (Erhan et al., 2009; Mordvintsev et al., 2015; Olah et al., 2017). While attribution/saliency methods attempt to explain how a network responds to an individual sample, feature visualizations attempt to explain the general sensitivity of a unit (e.g., a single channel of a convolutional layer) in a neural network. This is achieved by visualizing highly activating patterns through activation maximization. First introduced by Erhan et al. (2009), feature visualizations have continually been refined through better priors and regularization terms that improve their intuitive appeal (e.g., Yosinski et al., 2015; Mahendran & Vedaldi, 2016; Nguyen et al.,

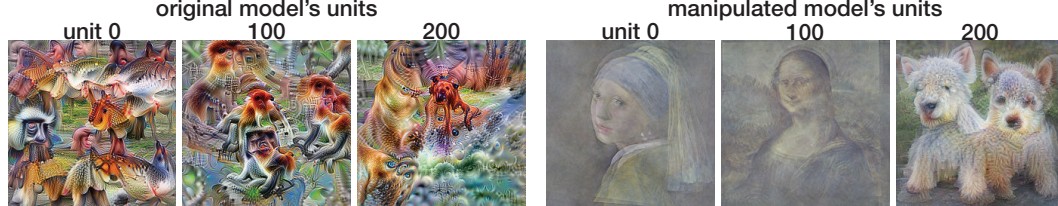

Figure 1: **Arbitrary feature visualizations.** Don't trust your eyes: Feature visualizations can be arbitrarily manipulated by embedding a fooling circuit in a network, which changes visualizations while maintaining the original network's ImageNet accuracy. **Left:** Original feature visualizations. **Right:** In a network with a fooling circuit, feature visualizations can be tricked into visualizing arbitrary patterns (e.g., Mona Lisa).

2016; Olah et al., 2017; Fel et al., 2023). Today, feature visualization methods underpin many of our intuitions about the inner workings of neural networks. They have been proposed as debugging tools (Nguyen et al., 2019), found applications in neuroscience (Walker et al., 2019; Bashivan et al., 2019; Ponce et al., 2019), and according to Olah et al. (2017), "to make neural networks interpretable, feature visualization stands out as one of the most promising and developed research directions." So what do we know about feature visualization's reliability? Despite its widespread use within the mechanistic interpretability community, relatively little: While they appear to provide some information, humans often struggle to make sense of those visualizations (Gale et al., 2020; Borowski et al., 2021; Zimmermann et al., 2021). Furthermore, we know that "by itself, feature visualization will never give a completely satisfactory understanding" (Olah et al., 2017), but we don't know to which degree we can trust or rely on them. After initial excitement, many areas of interpretability research have become more cautious and sceptical in general—but scepticism alone is not going to answer important questions such as: Can a method be fooled? How may we sanity-check its reliability? And under which circumstances can the method be guaranteed to be reliable? In this article we provide answers to those three questions:

1. **Adversarial perspective: Can feature visualizations be fooled?** We develop fooling circuits that trick feature visualizations into displaying arbitrary patterns or visualizations of unrelated units. Thus, feature visualizations can be deceived if one has access to the model (Section 2).

2. **Empirical perspective: How can we sanity-check feature visualizations?** We provide a simple sanity check and show that feature visualizations, which are widely used for mechanistic interpretability, are processed largely along different paths compared to natural images, casting doubt on their ability to explain how neural networks process natural images (Section 3).

3. **Theoretical perspective: Under which circumstances is feature visualization guaranteed to be reliable?** Our theory proves that this is only possible if we know a lot about the network already, and impossible if the network is a black-box (Section 4).

We do not mean to imply that feature visualizations per se are not a useful tool for analyzing hidden representations (they are, and it is important to know how individual parts of a neural network function). Instead, we hope that our investigations can help inspire the development of more reliable feature visualizations: a *synthesis* or new avenue.

## 2   ADVERSARIAL PERSPECTIVE: CAN FEATURE VISUALIZATIONS BE FOOLED?

One important requirement for interpretability is that the explanations are reliable. We use the following definition of unreliability: A visualization method is unreliable if one can change the visualizations of a unit without changing the unit's behavior on (relevant) test data. More formally, this can be expressed as: Let $\mathcal{U}$ denote the space of all units. A visualization method $m$ is unreliable if $\exists u, v \in \mathcal{U} : m(u) = m(v) \wedge \neg u \overset{bhv}{\sim} v$, where $\overset{bhv}{\sim}$ denotes an equivalence class of equal behavior.

To understand the reliability of feature visualizations, we start by actively deceiving visualizations. For this, we design two different fooling methods: a *fooling circuit* (Section 2.1) and *silent units* (Section 2.2). The motivation for this is twofold. Most importantly, if we can show that one can

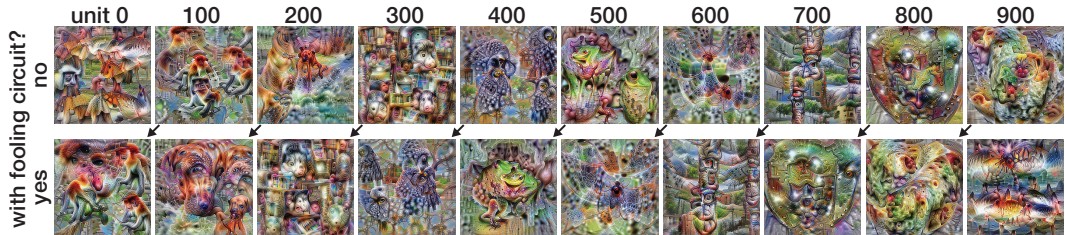

Figure 2: **Using a fooling circuit to arbitrarily permute visualizations. Top row:** Visualizations of the last-layer units in the original Inception-V1 model. **Bottom row:** After integrating a fooling circuit, units show an arbitrarily permuted visualization (here: offset by 100 indices).

actively fool feature visualizations, this provides a proof of concept by showing that it is possible to build networks where feature visualizations are completely independent of network behavior on natural images. This concept (different network behavior for natural images vs. feature visualizations) is later investigated for non-adversarial settings both empirically and theoretically. Furthermore, since feature visualizations have been proposed as model auditing tools (Brundage et al., 2020) that should be integrated "into the testbeds for AI applications" (Nguyen et al., 2019, p. 20), it is important to understand whether an adversary (i.e., someone with malicious intent) might be able to construct a model such that feature visualizations are manipulated. This corresponds to a **threat scenario** where the model itself can be arbitrarily changed while the interpretability technique (feature visualization) is kept fixed without control over hyperparameters or the random starting point. For example, a startup may be interested in hiding certain aspects of its model's behavior from a third-party (e.g. regulator) audit that uses feature visualizations. In this context, our demonstration of unreliability relates to a line of work on deceiving other interpretability methods (described in detail in Appendix A.1). We don't know whether such a scenario could become realistic in the future, but as we stated above the main motivation for this section is to provide a proof of concept by showing that one can build networks that deceive feature visualizations.

## 2.1 MANIPULATING FEATURE VISUALIZATIONS THROUGH A FOOLING CIRCUIT

Our first method to deceive feature visualizations is a *fooling circuit*. It can be embedded in a standard neural network architecture and changes how feature visualizations look without changing the behavior of the network on natural input. By circuit we mean a set of interconnected units carrying out a specific function (Pulvermüller et al., 2014; Olah et al., 2020). In the literature, the term unit either means a single convolutional channel in a convolutional layer or a single neuron $u$ in a fully-connected layer that computes $u(x) = \text{ReLU}(Wx + b)$. For the sake of introducing the fooling circuit, we use the latter definition. We start by taking a standard pre-trained neural network, Inception-V1 (Szegedy et al., 2015), and randomly pick a unit in the last layer (i.e., just before the softmax is applied). When visualizing this unit, denoted $F$, using the standard visualization method by Olah et al. (2017), we might see, for instance, feathers if this unit corresponds to class "feather" (given that the unit is picked from the last layer, the unit is class-selective since the network was trained to do object classification). The goal of the fooling circuit is to insert a new deceptive network unit $A$ that shows two different modes of behavior: If the network processes natural images, the unit should respond to feathers just like unit $F$, whereas if feature visualization is performed on the unit, the unit's visualization should depict something completely different, e.g., a donut. We achieve this by wiring six units with ReLU activation functions together as shown in Figure 4.

The fooling circuit has three layers with three, two and one unit(s) each. For the first layer consisting of three units, we start by taking an existing unit $D$ that responds to donuts and the existing unit that responds to feathers $F$. In the same layer, we attach a binary classifier unit $E$ that operates directly on the network's input. This classifier is not part of the original network. It is trained to tell whether the network's input is a natural image (in which case the classifier output is 1) or a synthetic feature visualization (output 0). Note that this binary classifier has a relatively simple task since feature visualizations start from random noise and end up at a synthetic visualization; thus they are clearly distinguishable from natural images at every step of the process (see Figure 3 for examples). The middle layer consists of two units ($B, C$) and the last layer of a single unit $A$ which will become the unit showing deceptive behavior by responding to natural feather images while feature-visualizing

feature visualization trajectory                    natural ImageNet validation images

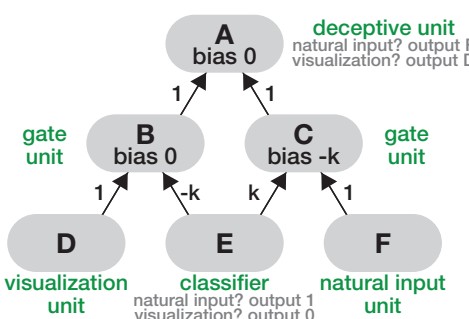

Figure 3: **Natural vs. synthetic distribution shift.** There is a clear distribution shift between feature visualizations (left) and natural images (right). This can be exploited by a classifier when building a fooling circuit. Visualizations at different steps in the optimization process for a randomly selected unit in the last layer of standard, unmanipulated Inception-V1; randomly selected ImageNet validation samples (excluding images containing faces).

Figure 4: **Fooling circuit.** This circuit consists of six units. Unit $A$ responds like unit $F$ for natural images, but the feature visualizations of $A$ are identical to the ones of $D$. This is achieved by a classifier unit ($E$) distinguishing between natural and visualization input, and two intermediate units with ReLU nonlinearities ($B$ and $C$) selectively suppressing information depending on the classifier's output. $k$ is an arbitrary large constant that ensures the gradient flows only through either the left or the right part of the circuit, not both, by pushing either $B$ or $C$'s pre-ReLU activation below zero.

a donut. The weights connecting the layers are set as follows: The activation of $A$ is identical to $B$ which is identical to $D$ (the donut unit) for feature visualizations; for natural input, the activation of $A$ is identical to $C$ which is identical to $F$ (the feather unit). To distinguish between those two cases, the classifier detects feature visualizations and depending on its output, either unit $B$ or unit $C$ has a zero activation since the classifier activation selectively suppresses the pre-ReLU activations of those units by pushing them below zero. Hence, their activations are exactly zero after the ReLU. As a result, inserting this fooling circuit creates a unit $A$ with donut feature visualizations yet when processing natural images, the unit ignores donuts and only responds to feathers.

Using this canonical fooling circuit, we can not only create a single deceptive unit but also simply replicate the circuit for each unit in the last layer of a model and thus, for instance, arbitrarily permute the visualizations of an entire layer as shown in Figure 2. We can even go so far as to design completely arbitrary visualizations through a small modification to the circuit: instead of connecting an existing unit (like the donut unit $D$), we can add a new unit $D'$ where the receptive field (a single convolutional filter) is hard-coded to an arbitrary pattern such as the Mona Lisa, as shown in Figure 1. Irrespective of how we manipulate the feature visualizations, the network still responds normally to natural input. This can be verified by checking the network's validation accuracy on ImageNet-1K, which only minimally changes when deceiving all visualizations in the last layer of Inception-V1 (top-1 accuracy changes from 69.146 % to 68.744 %; top-5 from 88.858 % to 88.330 %). The tiny drop in performance is a result of the binary classifier achieving slightly less-than-perfect accuracy ($99.49\%$ on a held-out test set) when distinguishing between natural input and visualization input. More experimental details are available in Appendix C.1.

Our fooling circuit shows that **it is possible to maintain essentially the same network behavior on natural input while drastically altering feature visualizations**. In Appendix B.1, we formalize this fooling circuit, and prove that it will always behave in the way we observe in our experiments. Given that this fooling circuit requires training a binary classifier, we next explore an alternative fooling method without this requirement.

## 2.2 MANIPULATING FEATURE VISUALIZATIONS BY LEVERAGING SILENT UNITS

Our second fooling method does not require a separate classifier but instead leverages orthogonal filters embedded in *silent units*; i.e., units that do not activate for the entire training set. We designed this method to show that fooling can easily be achieved in different ways and across different architectures. In order to demonstrate that different model families can be fooled, we here consider a

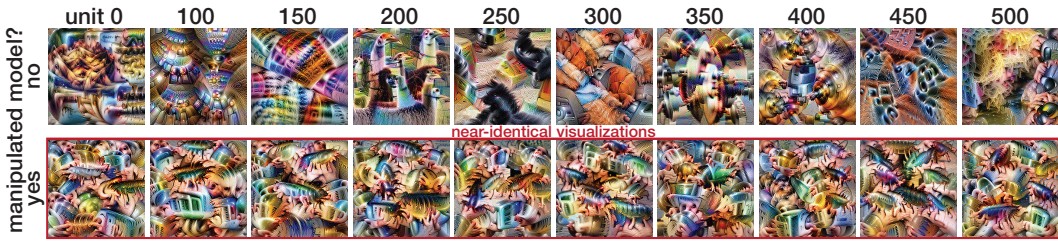

Figure 5: **Leveraging silent units to produce identical visualizations throughout a layer.** The top row shows feature visualizations for units of a layer (block 4-1, conv 2) in a standard, unmanipulated ResNet-50. For the bottom row, we manipulate the model such that the feature visualizations of all units become near-identical (indicated by the red box). Nevertheless, the units still perform the same computations as in the original model on natural input, as evident by an unchanged validation loss. This is achieved by leveraging orthogonal filters in silent units as described in Section 2.2.

different architecture (ResNet-50 (He et al., 2016) instead of Inception-V1) and a randomly selected intermediate layer instead of the last layer (but note that the approach is not specific to a certain architecture or layer). We replace a standard computational block,

$$y = \text{ReLU}(\text{BatchNorm}(\text{Conv}(x, \Theta), \Psi)), \tag{1}$$

where $\Theta$ and $\Psi$ are learned conv / batch norm parameters, with the manipulated computation block

$$\bar{y} = y + \Delta y, \text{ where } \Delta y = \text{ReLU}(\text{Conv}(x, \bar{\Theta}) + b). \tag{2}$$

Our goal is to set $\bar{\Theta}$ and $b$ such that $\bar{y} = y$ on natural images while the feature visualizations of $\bar{y}$ are not related to those of $y$ and instead dominated by $\Delta y$. Since feature visualizations usually lead to substantially larger activations than natural samples, we can exploit this property without requiring an explicit classifier like the fooling circuit did. Note that we do not change the original unit, i.e., how $y$ is computed; this means that $\Theta$ and $\Psi$ stay unchanged. Instead, we introduce a new unit and choose its two free parameters $\bar{\Theta}$ and $b$ such that $\Delta y = 0$ for natural inputs but $\Delta y \neq 0$ for feature visualizations. Specifically, we construct $\bar{\Theta}$ as a linear combination of the weight $\Theta$ used to compute $y$ and a sufficiently strong orthogonal perturbation $\Delta\Theta^\perp$; that is, $\bar{\Theta} = \alpha\Theta + \beta\Delta\Theta^\perp$, where $\alpha, \beta$ control the relative strength of the filter directions. By choosing a sufficiently negative bias $b$, we ensure that $\Delta y$ remains silent (no activation, ensuring that natural input is processed as before) unless $y$ is very strongly activated. Letting $\bar{y}_{\text{max. nat}}$ denote the maximal observed activation on natural input for the constructed filter $\bar{\Theta}$, we set $b = -\alpha/\beta\,\bar{y}_{\text{max. nat}}$. Since we empirically observe a large gap between activations reached by feature visualizations and natural input, we are able to steer the visualizations to (almost) arbitrarily chosen images. We demonstrate this by applying it to a ResNet-50 model trained on ImageNet. In Figure 5, all 512 units in a ResNet layer yield near-identical feature visualization. This has no impact on the overall behavior of the network: neither the top-1 nor the top-5 validation accuracy change at all. In summary, we developed two different methods that trick feature visualizations into showing arbitrary, permuted, or identical visualizations across a layer. This establishes that **feature visualizations can be arbitrarily manipulated** if one has access to the model.

## 3 EMPIRICAL PERSPECTIVE: HOW CAN WE SANITY-CHECK FEATURE VISUALIZATIONS?

In Section 2 we have seen that feature visualizations can be fooled under adversarial circumstances. However, it is safe to say that in most cases we do not expect an adversary to manipulate a network. Therefore, a much more pressing question is: How can we check whether feature visualizations work as intended under normal circumstances, i.e., for standard, unmanipulated networks? In the context of saliency methods, sanity checks have proven highly valuable for investigating method reliability (Adebayo et al., 2018). We here provide an empirical sanity check for feature visualizations.

The core idea is simple: Feature visualizations are designed to explain how neural networks process natural input. This means that once they are generated, good visualizations should be processed along a similar path as natural images—which is an aspect that we can verify empirically. Let's say

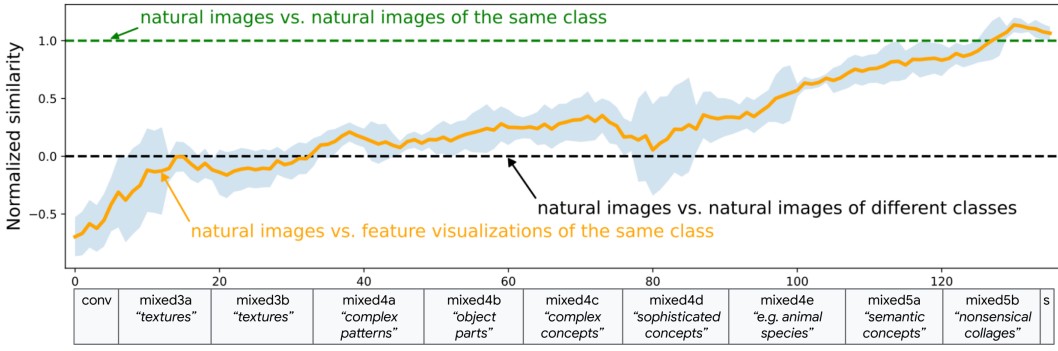

Figure 6: **Sanity check: Feature visualizations are processed differently than natural images.** Feature visualizations are designed to explain how neural networks process natural input—but do feature visualizations for a certain class actually activate similar units as natural input from this class? We measure the similarity of a layer's activations caused by natural images and feature visualizations across layers. Throughout the first two thirds of Inception-V1 layers, activations of natural images have roughly as little similarity to same-class visualizations as they have to completely arbitrary images of different classes. In the last third of the network, similarity increases. Layer annotations (e.g., "textures" / "object parts") are from Olah et al. (2017).

we take a unit from the last layer of a network, a unit for which we know that it responds to "cat" images. If we run feature visualization on this unit and feed the resulting visualization through the network, a good "cat" visualization should show typical cat features and thus activate very similar units as natural cat images. Generally speaking, we expect images from the same class to be processed along a similar path because they share certain features. For instance, all airplanes have wings, and all cats have paws. Some features are shared across classes (e.g., both cat and airplane images may contain a blue sky), and some are more class-specific (airplane: wings, cats: paws). If a neural network contains units that respond to certain features—for instance, one unit responds to paws and a different unit to wings—then natural images of the same class should, to a certain degree, activate similar units; and those units should also become activated when processing a good feature visualization. Conceptually, this approach is motivated by the *fooling circuit* from Section 2.1, where the circuit leads to feature visualizations being disconnected from network behavior on natural input by using different paths for different inputs. Hence, we can proceed by analyzing the following three properties for each layer of a standard network:

- How similarly are natural images from the same class processed (e.g., one cat image vs. another cat image)? This serves as the upper bound: the maximal similarity we can hope to capture with a good feature visualization.

- How similarly are natural images from different classes processed (e.g., a cat image vs. an airplane image)? This serves as a lower bound: the baseline similarity we can expect from processing completely unrelated input.

- How similarly are natural images from a class vs. feature visualizations for the same class processed? This is an indication of the reliability of the feature visualization method.

For the sake of this sanity check, we focus on feature visualizations (Olah et al., 2017) for the last layer of a standard network, Inception-V1. The last layer is a perfect choice for this kind of analysis since in contrast to hidden layers, the units in the last layer have perfectly well-known ground truth selectivity: each unit is selective for one class. In terms of measuring similarity between activations of input $x_i$ and input $x_j$ for a network $f$ in layer $l$, we compute $\Gamma(f_l(x_i), f_l(x_j))$ where $\Gamma$ could be any similarity metric. We here use Spearman's rank order correlation but our findings are not limited to this metric—other choices such as Cosine Similarity or Pearson correlation lead to the same results, as can be seen in Appendix C.3. Using this similarity metric, we can compare whether images of a class are processed similarly to feature visualizations for the same class throughout the network. If so, they should activate roughly the same units in preceding layers (similar activations → high correlation). If they are processed along arbitrary independent paths instead, we would obtain zero correlation. In Figure 6, we plot the results of this analysis, normalized such that a value of 1 corresponds to the Spearman similarity obtained by comparing natural images of the same class

(airplanes vs. airplanes, cats vs. cats), and 0 corresponds to the similarity that is obtained from comparing images of one class against images of a different class (airplanes vs. cats etc.,). The results are averaged across classes; raw data and additional information can be found in Appendix C.3.

As can be seen in Figure 6, last-layer feature visualizations are processed differently from natural images throughout most of the network. If they would be processed along the same path, similarity would need to be high across all layers. Later layers have a higher correlation, but that does not mean that the activations are resulting from the same paths. In many earlier and mid-level layers, the activations of, say, cat images are as similar to activations of cat visualizations as they are to activations of flower, airplane or pizza images. While it would be fine for a feature visualization to show different low-level features compared to natural images, any visualization that seeks to explain how natural input is processed should capture similarities in mid- and high-level layers that are described by Olah et al. (2017) as responding to "object parts" and "complex/sophisticated concepts". This means that throughout most of the network, the investigated feature visualization does not pass the sanity check: **processing along different paths casts doubt on the ability of feature visualizations to explain how standard neural networks process natural images**. Looking ahead, we hope that the similarity sanity check we introduce here facilitates rigorous, quantitative evaluation of feature visualizations and guides researchers in designing more reliable feature visualization methods.

## 4 THEORETICAL PERSPECTIVE: UNDER WHICH CIRCUMSTANCES IS FEATURE VISUALIZATION GUARANTEED TO BE RELIABLE?

A natural question arising from our experiments is whether the limitations of feature visualization that we have shown experimentally are always present, or if they can be avoided in certain situations. Towards addressing this, we now ask: When are feature visualizations guaranteed to be reliable, i.e., guaranteed to produce results that can be relied upon? Feature visualizations are expected to help us "*answer what the network detects*" (Olah et al., 2018), "*understand what a model is really looking for*" (Olah et al., 2017), and "*understand the nature of the functions learned by the network*" (Erhan et al., 2009). When formalizing these statements, two aspects need to be considered. First, the structure of functions to be visualized. The current literature does not place assumptions on the function—it could be any unit in a "*black-box*" (e.g., Heinrich et al., 2019; Nguyen et al., 2019) neural network. Second, we need to characterize which aspects of the function feature visualizations promise to help understand. The existing literature (quotes above and in Appendix A.2) broadly claims that feature visualizations are useful for 'understanding' a function $f$ (such as a unit in a neural network). If that is indeed the case, then a user should be able to use feature visualizations to make meaningful predictions about the behavior of $f$ on some input $x$. Our theory quantifies this by assessing whether it is possible to predict $f(x)$ for any network input $x$ based on feature visualizations. We investigate three different settings (see Table 1): exact prediction, approximate prediction up to an error $\varepsilon$, and predicting at least whether $f(x)$ is closer to the min- or maximum of $f$. If none of these can be predicted, then feature visualizations cannot be said to have provided us with any meaningful understanding of the $f$. We investigate these three scenarios for twelve different function classes, ranging from general black-box functions (no assumptions about the function) to more restrictive settings (e.g., assuming $f$ to be convex).

Conceptually, our theory is based on the insight that feature visualization based on activation maximization seeks to synthesize a highly activating image, which corresponds to finding the $\arg\max$ of $f$—an insight that might seem trivial. Paradoxically, it is well-known that it is impossible to conclude much, if anything, about an unconstrained function from its $\arg\max$. Yet, feature visualizations are purported to help us understand what black-box functions (e.g., neural network units) detect. To resolve this paradox, we can impose stronger assumptions on the function, or lower our expectations by considering successively weaker notions of understanding, such as instead of asking whether a feature visualization can help predict $f(x)$ simply asking whether it can tell us at least whether the activation for a new test image $x$ will be closer to the maximum or the minimum of the function. In this section, we explore both directions, and show that **even strong assumptions on the function $f$ are *insufficient* to guarantee that feature visualizations are reliable for understanding $f$, even for very weak notions of understanding.** The core results of our theory are summarized in Table 1; exact definitions for each function class as well as proofs are in Appendix B.

Table 1: **Theory overview.** Feature visualization aims to help understand a function $f$ (e.g., a unit in a network). While understanding is an imprecise term, it can be formalized: Given $f$ and its arg max $x_{\max}$ and arg min $x_{\min}$ (approximated by feature visualization), how well can we predict $f(x)$ for new values of $x$? We show that this is impossible if $f$ is a black-box. Instead, to make meaningful predictions, we need strong additional knowledge about $f$.

| | | | Given feature visualization for a function $f$ and an input $x$, can we reliably predict... | | |
| | | | $f(x)$? | $f(x)$ $\varepsilon$-approx.? | if $f(x)$ is closer to $f_{\max}$ or $f_{\min}$? |
|---|---|---|---|---|---|
| Stronger assumptions about $f$ | black-box | $\mathcal{F}$ | No | No | No |
| | neural network (NN) | $\mathcal{F}_{\text{NN}}$ | No | No | No |
| | NN trained with ERM | $\mathcal{F}_{\text{ERM}}$ | No | No | No |
| | $L-$Lipschitz (known $L$) | $\mathcal{F}_{\text{Lip}}^{L}$ | No | No | Only for small $L$ |
| | piecewise affine | $\mathcal{F}_{\text{PAff}}$ | No | No | No |
| | monotonic | $\mathcal{F}_{\text{Mono}}$ | No | No | No |
| | convex | $\mathcal{F}_{\text{Convx}}$ | No | No | No |
| | affine (input dim. $> 1$) | $\mathcal{F}_{\text{Aff}}^{d>1}$ | No | No | No |
| | affine (input dim. $= 1$) | $\mathcal{F}_{\text{Aff}}^{d=1}$ | Yes | Yes | Yes |
| | constant | $\mathcal{F}_{\text{Const}}$ | Yes | Yes | Yes |

**Notation and definitions.** We denote the indicator function of a Boolean expression $E$ as $\mathbf{1}_E$, which is 1 if $E(x)$ is true and 0 otherwise. Let $d$ denote the input dimensionality (e.g., number of pixels and channels), $\mathcal{I} = [0,1]^d$ the input space, and $\mathcal{F} = \{\mathcal{I} \to [0,1]\}$ the set of all functions from inputs to scalar, bounded activations.[1] A *maximally activating feature visualization* is the map from $\mathcal{F}$ to $\mathcal{I}^2 \times [0,1]^2$ that returns a function's arg min, arg max, and values at these two points, which we denote by $\Phi_{\min\max}(f) = (\arg\min_{x\in\mathcal{I}} f(x), \arg\max_{x\in\mathcal{I}} f(x), \min_{x\in\mathcal{I}} f(x), \max_{x\in\mathcal{I}} f(x))$. When $f$ is clear from context, we write $\Phi_{\min\max} = (x_{\min}, x_{\max}, f_{\min}, f_{\max})$ for brevity. We assess the reliability of a feature visualization by how well it can be used to predict $f(x)$ at new inputs $x$. To make such a prediction, the user must *decode* feature visualization into useful information. We denote a *feature visualization decoder* as a map $D \in \mathcal{D} = \{\mathcal{I}^2 \times [0,1]^2 \to \mathcal{F}\}$. Our results do not rely on the structure of $D$ in any way. Rather, "No" in Table 1 means that for *every* $D$ the assumptions are insufficient to guarantee accurate prediction of $f$.

### 4.1 MAIN THEORETICAL RESULTS

Throughout, we measure the accuracy of predicting $f$ using $\|\cdot\|_\infty$. This is primarily for convenience; the equivalence of $L_p$ norms on bounded, finite-dimensional spaces implies we could prove the same results with $\|\cdot\|_p$ for any $p$ at the expense of dimension-dependent constants. This captures many cases of interest: $\|\cdot\|_p$ for $p \in \{1, 2\}$ is the standard measure of accuracy for regression, and for $f$ that outputs bounded probabilities, the logistic loss is equivalent to $\|\cdot\|_2$ up to constants.

First, we note that the boundedness of $f$ implies a trivial ability to predict $f(x)$.

**Proposition 1.** *There exists $D \in \mathcal{D}$ such that for all $f \in \mathcal{F}$,*

$$\left\| f - D(\Phi_{\min\max}(f)) \right\|_\infty \leq \frac{f_{\max} - f_{\min}}{2}. \tag{3}$$

This means that for any function $f$, a user can take the feature visualization $\Phi_{\min\max}(f)$ and apply a specific decoder (the constant function taking value halfway between $f_{\min}$ and $f_{\max}$) to predict $f(x)$ for *any* new $x$. If the user imposed assumptions on $f$, one might conjecture that a clever choice of decoder could lead to a better prediction of $f(x)$. Our first main result shows that this is impossible even for strong assumptions.

---

[1] For example, class probabilities or normalized activations of a bounded unit in a neural network.

**Theorem 1.** *For all $\mathcal{G} \in \{\mathcal{F}, \mathcal{F}_{\mathrm{NN}}, \mathcal{F}_{\mathrm{ERM}}, \mathcal{F}_{\mathrm{PAff}}, \mathcal{F}_{\mathrm{Mono}}\}$, $D \in \mathcal{D}$, and $f \in \mathcal{G}$, there exists $f' \in \mathcal{G}$ such that $\Phi_{\min\max}(f) = \Phi_{\min\max}(f')$ and*

$$\left\| f' - D(\Phi_{\min\max}(f')) \right\|_\infty \geq \frac{f'_{\max} - f'_{\min}}{2}. \tag{4}$$

Consider a user who knows that the unit to visualize is piecewise affine ($f \in \mathcal{F}_{\mathrm{PAff}}$). Using this knowledge, they hope to predict $f$ by applying some decoder to the visualization $\Phi_{\min\max}(f)$. However, for every $f$, there is always another $f'$ that satisfies the user's knowledge ($f' \in \mathcal{F}_{\mathrm{PAff}}$) and has $\Phi_{\min\max}(f) = \Phi_{\min\max}(f')$. Therefore, without further information it is impossible to distinguish between the case when the true function is $f$ and when it is $f'$, regardless of how refined the decoder is. Theorem 1 says that $f$ and $f'$ are sufficiently different, and thus, the user will do poorly at predicting at least one of them; that is, the user does not improve on the uninformative predictive ability prescribed by Proposition 1. This implies No for the first two columns in Table 1. A similar result can be shown for $\mathcal{F}_{\mathrm{Convx}}$ and $\mathcal{F}^L_{\mathrm{Lip}}$ as shown in Theorem 3 and Theorem 4, accordingly.

Our second result is an analogous negative result for predicting whether $f(x)$ is closer to $f_{\max}$ or $f_{\min}$, implying No for the third column in Table 1. To state it, for any $f \in \mathcal{F}$ we define $m_f = (f_{\max} + f_{\min})/2$; note that $f(x)$ is closer to $f_{\max}$ iff $f(x) > m_f$.

**Theorem 2.** *For all $\mathcal{G} \in \{\mathcal{F}, \mathcal{F}_{\mathrm{NN}}, \mathcal{F}_{\mathrm{ERM}}, \mathcal{F}_{\mathrm{PAff}}, \mathcal{F}_{\mathrm{Mono}}, \mathcal{F}_{\mathrm{Convx}}\}$, $D \in \mathcal{D}$, and $f \in \mathcal{G}$, there exists $f' \in \mathcal{G}$ such that $\Phi_{\min\max}(f) = \Phi_{\min\max}(f')$ and*

$$\left\| \mathbf{1}_{f' > m_{f'}} - \mathbf{1}_{D(\Phi_{\min\max}(f')) > m_{f'}} \right\|_\infty \geq \mathbf{1}_{f_{\max} \neq f_{\min}}. \tag{5}$$

The LHS of Eq. (5) quantifies "Can the user tell if $f'(x)$ is closer to $f'_{\max}$ or $f'_{\min}$?" Since indicator functions are bounded in $[0,1]$, the LHS is trivially bounded above by 1. Again consider the user who knows $f \in \mathcal{F}_{\mathrm{PAff}}$. Theorem 2 says that for any $f$—unless $f$ also happens to be constant (i.e., $f_{\max} = f_{\min}$)—there is always some $f' \in \mathcal{F}_{\mathrm{PAff}}$ that is indistinguishable from $f$ to the user *and* sufficiently different from $f$ so that the user cannot reliably tell if $f'(x)$ is closer to $f'_{\max}$ or $f'_{\min}$ (i.e., the RHS is also 1). The same result can be shown for $\mathcal{F}^L_{\mathrm{Lip}}$ with dependence on $L$ (Theorem 5).

For an analogous analysis and further results for more function classes, see Theorems 6 to 8. In summary, we prove that **without additional assumptions about a function, it is impossible to guarantee that standard feature visualizations can be used to understand (i.e., predict) many types of functions, including black boxes, neural networks, and even convex functions**. That said, if strong additional knowledge is available, for instance, if the function is known to be affine with low input dimensionality, then feature visualizations are provably reliable. In line with other work (Srinivas & Fleuret, 2019; Bilodeau et al., 2022; Fokkema et al., 2022; Han et al., 2022), this marks a departure from the conventional concept of black-box interpretability and suggests that more knowledge about the function—for instance, enforced through architectural primitives—are necessary to ensure feature visualization reliability.

## 5 ARE MORE LINEAR UNITS EASIER TO INTERPRET?

The theory above makes a prediction: the simpler a function is, the easier it should be to interpret given a feature visualization. We here empirically test this prediction. As a measure of simplicity, we use *path linearity*: based on a highly activating natural image (start point), we perform feature visualization to arrive at a highly activating optimized image (end point). We then analyze how much the optimization path deviates from linearity by measuring the angle between gradients of the first $n$ steps of the path. This metric (across many such paths) is then compared to human experimental data from Zimmermann et al. (2023) for the same units, who measured how well humans can interpret different units in Inception-V1. Intriguingly, we find a significant correlation (Spearman's $r = -.36$, $p = .001$) between this human interpretability score and our path linearity measure (lower angle means higher interpretability) especially for the beginning of the trajectory (e.g., $n = 2$); a plot can be found in Appendix C.4. Overall, we interpret this as preliminary evidence in favor of the hypothesis that more linear units, at least at the beginning of the optimization trajectory, might be easier to interpret. An interesting direction for future work would be to enforce higher degrees of

linearity, for instance through regularization or the architecture. This is one example of how our theory might be used to develop hypotheses for better feature visualizations.

## 6  CONCLUSION

Feature visualizations based on activation maximization are a widely used tool within the mechanistic interpretability community. We here asked whether feature visualizations are reliable, i.e., whether we can trust/rely on their results. Our work has the following practical implications:

1. **Adversarial perspective: Feature visualizations can be arbitrarily manipulated/fooled.** Thus, contrary to calls in the literature, feature visualizations are not a reliable tool for model auditing by a third party that did not train the model itself (e.g., a regulator).

2. **Empirical perspective: A sanity check shows that feature visualizations are processed very differently from natural images; they currently do not explain how natural input is processed.** We therefore recommend: (a) using feature visualization for *exploratory* but not for *confirmatory* use cases; (b) consistent with the recommendation by Olah et al. (2018; 2020), always combining visualizations with additional methods including dataset samples and (c) when proposing a new feature visualization method, measuring whether it reflects how natural input is processed throughout the network using the quantitative sanity check we introduced.

3. **Theoretical perspective: Feature visualization based on activation maximization can only be guaranteed to be reliable if we know a lot about the network already; it's impossible if the network is a black-box.** This challenges the concept of post-hoc interpretability methods: explaining completely black-box systems may sometimes be more than we can hope for.

We believe that developing novel, more reliable feature visualizations is a challenging and important direction for future work.Given that our theory proves that visualizing black-box systems via feature visualizations based on activation maximization (i.e., the current dominant paradigm) cannot be guaranteed to be reliable without making strong assumptions about the system, we see two potential avenues: either radically deviating from activation maximization, or making much stronger assumptions on the network (e.g., stronger linearity as explored in Section 5). In either case, it is important to understand the problem first before we can solve it—in fact, developing solutions may well be a multi-year effort. This article aims to convince readers that there is indeed an important problem, proposes a sanity check, develops a theoretical framework for reliability guarantees, and seeks to motivate future work on solutions.

### REPRODUCIBILITY STATEMENT

Code to replicate experiments from this paper is available from the supplementary material and will be open-sourced on GitHub. The proofs for our theory can be found in Appendix B. Method details beyond the descriptions from the main text, including the choice of hyperparameters, are available from our extensive 18-page appendix as well (Appendix C). There are no special compute requirements (e.g., we do not train large models). Information pertaining to code libraries and feature visualization details can be found in Appendix C.2. We added a table of contents at the beginning of the appendix section to facilitate accessibility.

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

# Appendix

## Table of Contents

## A    LITERATURE

### A.1    RELATED WORK ON DECEIVING INTERPRETABILITY METHODS

Our experiments from Section 2 serve two purposes. First, we provide a *proof of concept* that it is possible to develop networks with arbitrary or misleading visualizations. Second, feature visualizations have been proposed as model auditing tools (Brundage et al., 2020) that should be integrated "into the testbeds for AI applications" (Nguyen et al., 2019, p. 20). Our work demonstrates the first "interpretability circumvention method" (term by Sharkey, 2022) for feature visualization, which corresponds to a well-known *attack scenario* where an entity wants to hide certain network behavior (e.g., to fool a third-party model audit or regulator). For instance, the literature considers scenarios where a model bias is discovered (e.g., a model exploits protected attributes like gender for classification), but since removing this bias decreases model performance, there is an incentive to hide the bias instead (Heo et al., 2019; Anders et al., 2020; Shahin Shamsabadi et al., 2022) without compromising model performance. Adapting models to maintain their behavior on standard input while showing malicious behavior under adversarial circumstances is known under various names: *fairwashing* if the goal is to hide model bias (Anders et al., 2020; Aïvodji et al., 2019), *model backdooring* or *weight poisoning* (Chen et al., 2017; Gu et al., 2017; Adi et al., 2018) (applied to

saliency maps by (Fang & Choromanska, 2022; Noppel et al., 2022)), *data poisoning* (Goldblum et al., 2022) if the change in model weights is achieved through interfering with the training data (explored by Baniecki et al. (2023) in the context of explanation methods), *adversarial model manipulation* to fool saliency maps (Heo et al., 2019), and *scaffolding* for fooling LIME and SHAP (Slack et al., 2020). Thus, while we are the first to successfully deceive feature visualizations in this manner, the scenario of adapting a model to deceive an interpretability method has a rich history. A complementary approach is proposed by Nanfack et al. (2023), which changes highly activating dataset samples without changing feature visualizations; finally, Sabour et al. (2015) demonstrated that hidden representations of neural networks can be adversarially manipulated, fooling the early visualization method by Mahendran & Vedaldi (2015).

## A.2 LITERATURE EXPECTATIONS ABOUT FEATURE VISUALIZATION

This short section provides a few expectations/hopes that are presented in the literature when it comes to feature visualizations.

Original activation maximization paper by Erhan et al. (2009):

- "a pattern to which the unit is responding maximally could be a good first-order representation of what a unit is doing"

- "It is perhaps unrealistic to expect that as we scale the datasets to larger and larger images, one could still find a simple representation of a higher layer unit."

- "we hope that such visualization techniques can help understand the nature of the functions learned by the network"

More recent literature:

- "Feature visualization allows us to see how GoogLeNet, trained on the ImageNet dataset, builds up its understanding of images over many layers" (Olah et al., 2017)

- "Feature visualization answers questions about what a network—or parts of a network—are looking for by generating examples." (Olah et al., 2017)

- "If we want to find out what kind of input would cause a certain behavior—whether that's an internal neuron firing or the final output behavior—we can use derivatives to iteratively tweak the input towards that goal" (Olah et al., 2017)

- "optimization approach can be a powerful way to understand what a model is really looking for, because it separates the things causing behavior from things that merely correlate with the causes". "Optimization isolates the causes of behavior from mere correlations." (Olah et al., 2017)

- "In the quest to make neural networks interpretable, feature visualization stands out as one of the most promising and developed research directions. By itself, feature visualization will never give a completely satisfactory understanding. We see it as one of the fundamental building blocks that, combined with additional tools, will empower humans to understand these systems." (Olah et al., 2017)

- "To make a semantic dictionary, we pair every neuron activation with a visualization of that neuron and sort them by the magnitude of the activation."; "Semantic dictionaries give us a fine-grained look at an activation: what does each single neuron detect?" (Olah et al., 2018)

- "Feature visualization helps us answer what the network detects" (Olah et al., 2018)

- "The behavior of a CNN can be visualized by sampling image patches that maximize activation of hidden units [...], or by using variants of backpropagation to identify or generate salient image features" (Bau et al., 2017)

- "Activation maximization techniques enable us to shine light into the black-box neural networks." (Nguyen et al., 2019)

Critical voices:

- "While these methods may be useful for building intuition, they can also encourage three potentially misleading assumptions: that the visualization is representative of the neuron's behavior; that the neuron is responsible for a clearly delineated portion of the task or the network's behavior; and that the neuron's behavior is representative of the network's behavior." (Leavitt & Morcos, 2020)

- "synthetic images from a popular feature visualization method are significantly less informative for assessing CNN activations than natural images" (Borowski et al., 2021)

- "[We] find no evidence that a widely-used feature visualization method provides humans with better 'causal understanding' of unit activations than simple alternative visualizations" (Zimmermann et al., 2021)

- "Neural networks often contain 'polysemantic neurons' that respond to multiple unrelated inputs." (Olah et al., 2020)

- "Units similar to those [hand-picked units] may be the exception rather than the rule, and it is unclear whether they are essential to the functionality of the network. For example, meaningful selectivities could reside in linear combinations of units rather than in single units, with weak distributed activities encoding essential information." (Kriegeskorte, 2015)

### A.3    RELATIONSHIP TO HIGHLY ACTIVATING NATURAL SAMPLES

In the interpretability community, visualizing highly activating natural samples for certain units is often done either alongside or instead of feature visualizations (Olah et al., 2017; Borowski et al., 2021). A natural question to ask is whether our results would apply to highly activating natural samples, too. Since this paper covers three perspectives we have three answers to this question:

From the *adversarial perspective*, we could easily use our method from Section 2.2 to build a network where the top $k$ natural images do not correspond to what the unit is usually selective for. This can be achieved by setting the bias parameter $b$ to a smaller value such that it would only suppress activations up to the, say, 95th percentile of natural input. A recent paper specifically looked into manipulating the top-k activating images (Nanfack et al., 2023).

From the *empirical perspective*, highly activating natural images would pass the sanity check since highly activating natural images are, by definition, natural images that highly activate a unit and they would thus be processed like other natural images.

From the *theoretical perspective*, our impossibility results can be extended to many variations on using the argmax for feature visualization, including using the most activating dataset samples as explanations. Essentially, as long as the feature visualization method does not narrow down the function space too much, our results will apply. It is easy to see that two simple (e.g., piecewise linear) functions could have the same 5 (or 10, etc.) local (arg)maxima and yet behave very differently even quite near these local maxima, and hence our theorems could be extended.

Thus in summary, highly activating natural images would pass our sanity check but it would still be possible to construct networks that show misleading highly activating natural images, which is a case that is covered by the theory.

## B    PROOFS AND THEORY DETAILS

### B.1    PROOF FOR FOOLING CIRCUIT (SECTION 2.1)

**Lemma 1.** *Let $k > 0$, $A : \mathbb{R}_+ \times \mathbb{R}_+ \to \mathbb{R}$ and $B, C : \mathbb{R}_+ \times \{0, 1\} \to \mathbb{R}_+$ with*

$$A(x, y) = x + y$$
$$B(x, z) = \max(0, x - kz)$$
$$C(x, z) = \max(0, x + kz - k)$$

*be computations represented by a sub-graph of a neural network. Denote the combination of these computations as $N : \mathbb{R}_+ \times \mathbb{R}_+ \times \{0, 1\}$ with*

$$N(x, y, z) = A(B(x, z), C(y, z)).$$

*Then it holds that*

$$\forall x, y \in \mathbb{R}_+ : k \geq \max(x, y) \implies \begin{cases} N(x, y, 0) = x \\ N(x, y, 1) = y \end{cases}.$$

*Proof of Lemma 1.* First, consider the case where the binary input of $N$ is 0; that is, $z = 0$. Then, since $k \geq y$,

$$\begin{aligned} B(x, 0) &= \max(0, x) = x \\ C(y, 0) &= \max(0, y - k) = 0, \end{aligned} \tag{6}$$

so $N(x, y, 0) = A(B(x, 0), C(y, 0)) = A(x, 0) = x$.

Analogously consider $z = 1$. Then, since $k \geq x$,

$$\begin{aligned} B(x, 1) &= \max(0, x - k) = 0 \\ C(y, 1) &= \max(0, y) = y, \end{aligned} \tag{7}$$

so $N(x, y, 1) = A(B(x, 1), C(y, 1)) = A(0, y) = y$, completing the proof. □

**Lemma 2.** *Let $\mathcal{X}$ denote the space of all possible inputs (e.g., all images), $\mathcal{D}$ some distribution on $\mathcal{X}$ (e.g., ImageNet). Let $D, F : \mathcal{X} \to \mathbb{R}_+$ represent the full computation of a unit in the original and in the tinkered network, respectively, that can be bounded on their domain. For an arbitrary algorithm $\mathrm{Opt} : \{\mathcal{X} \to \mathbb{R}\} \times \mathcal{X} \to \mathcal{X}$ and distribution $\pi_0$ on $\mathcal{X}$ define the following sequence of random variables: $\forall n > 0 : X_{n+1} = \mathrm{Opt}(D, X_n)$ and $X_0 \sim \pi_0$. Denote the distribution over $\mathcal{X}$ induced by this process $\pi$. If $\mathcal{D}$ and $\pi$ have disjoint support, then there exists a neural network implementing a function $N : \mathcal{X} \to \mathbb{R}$ such that*

$$\mathbb{P}_{\mathbf{x} \sim \pi}[N(\mathbf{x}) = D(\mathbf{x})] = 1 \ \text{and} \ \mathbb{P}_{\mathbf{x} \sim \mathcal{D}}[N(\mathbf{x}) = F(\mathbf{x})] = 1.$$

*Proof of Lemma 2.* As $\mathcal{D}$ and $\pi$ have disjoint support this means that there exists a function $E : \mathbb{R} \to \{0, 1\}$ such that

$$\mathbb{P}_{\mathbf{x} \sim \pi}[E(\mathbf{x}) = 0] = 1 \ \text{and} \ \mathbb{P}_{\mathbf{x} \sim \mathcal{D}}[E(\mathbf{x}) = 1] = 1. \tag{8}$$

Let $k = \max(\max_{\mathbf{x} \in \mathcal{D}} D(\mathbf{x}), \max_{\mathbf{x} \in \pi} D(\mathbf{x}))$, which exists as both $D$ and $F$ are bounded. In line with Lemma 1, we construct $N$ as $N(\mathbf{x}) = A(B(D(\mathbf{x}), E(\mathbf{x})), C(F(\mathbf{x}), \neg E(\mathbf{x})))$. Per the universal approximation theorem (Hornik et al., 1989), there exists a neural network implementing the assumed function $E$. As all other computations (i.e., $A, B, C, D, F$) are implemented by a neural network, we can conclude that the constructed function $N$ can also be implemented by a neural network.

Applying Lemma 1 and Eq. (8) directly yields

$$\mathbb{P}_{\mathbf{x} \sim \pi}[N(\mathbf{x}) = D(\mathbf{x})] = 1 \ \text{and} \ \mathbb{P}_{\mathbf{x} \sim \mathcal{D}}[N(\mathbf{x}) = F(\mathbf{x})] = 1, \tag{9}$$

concluding the proof. □

**Remark 1.** *In the case of feature visualizations, the assumption that $\pi$ and $\mathcal{D}$ have disjoint support is plausible as demonstrated empirically in Section 2.1; this can also be visually appreciated from looking at Figure 3 showing a visualization trajectory which at no point resembles natural images.*
◁

### B.2 DETAILS ON INTERPRETATION OF TABLE 1

First, we elaborate on what No and Yes mean in Table 1.

The weakest form of answering No would be to find a *single* function $f$ where feature visualization cannot be used to predict $f$. At the other extreme, one could hope to show that feature visualization cannot be used to predict $f$ *for all* $f$. Unfortunately, this is trivially impossible to show: for every distinct value the feature visualization can take, one could pick a function $f$ that agrees with this

visualization (e.g., has the same $\arg\max$) and use this as the prediction. In light of this, we prove the next strongest impossibility result. When the answer is No, we show that *for all*[2] functions $f$ (except for a handful of corner cases, like constant functions), there exists another function $f'$ that gets the exact same feature visualization as $f$ yet cannot be accurately predicted. Similarly, for the cells with **extra assumptions in orange**, this means that the answer is No (as defined in the previous sentence) *unless these extra assumptions are satisfied*.

We measure predictive accuracy using the sup norm for simplicity, but our results could be extended to any other strictly convex loss. This is essentially the strongest result one could hope for: by the intermediate value theorem, any continuous $f$ must take every value in between $f_{\min}$ and $f_{\max}$, and hence it is impossible to prove that $f(x)$ can't be recovered *for every* $x$. On the contrary, for the cells where the answer is Yes, we can actually prove something much stronger than the converse of No: we prove that $f(x)$ can be predicted *for all* $x$ and *for all* $f$. This hints at the necessity of such strong assumptions. Either the function class is so simple that feature visualization reveals everything about every function, or feature visualization reveals hardly anything about any function.

To find the precise results that correspond to each cell of Table 1, see Table 2.

Table 2: Theoretical results corresponding to each cell of Table 1.

| | | Given feature visualization for a function $f$ and an input $x$, can we reliably predict... | | |
| --- | --- | --- | --- | --- |
| | | $f(x)$? | $f(x)$ $\varepsilon$-approx.? | if $f(x)$ is closer to $f_{\max}$ or $f_{\min}$? |
| | black-box | $\mathcal{F}$  Theorem 1 | Theorem 1 | Theorem 2 |
| | neural network (NN) | $\mathcal{F}_{\text{NN}}$  Theorem 1 | Theorem 1 | Theorem 2 |
| | NN trained with ERM | $\mathcal{F}_{\text{ERM}}$  Theorem 1 | Theorem 1 | Theorem 2 |
| | $L-$Lipschitz (known $L$) | $\mathcal{F}_{\text{Lip}}^{L}$  Theorem 4 | Theorem 4 | Theorem 5 |
| | piecewise affine | $\mathcal{F}_{\text{PAff}}$  Theorem 1 | Theorem 1 | Theorem 2 |
| | monotonic | $\mathcal{F}_{\text{Mono}}$  Theorem 1 | Theorem 1 | Theorem 2 |
| | convex | $\mathcal{F}_{\text{Convx}}$  Theorem 3 | Theorem 3 | Theorem 2 |
| | affine (input dim. $> 1$) | $\mathcal{F}_{\text{Aff}}^{d>1}$  Theorem 6 | Theorem 6 | Theorem 7 |
| | affine (input dim. $= 1$) | $\mathcal{F}_{\text{Aff}}^{d=1}$  Theorem 8 | Theorem 8 | Theorem 8 |
| | constant | $\mathcal{F}_{\text{Const}}$  Theorem 8 | Theorem 8 | Theorem 8 |

(left vertical label: Stronger assumptions about $f$)

---

[2]Affine functions are the only exception, since there are more cases where an affine $f$ can be exactly recovered from feature visualization. See Theorem 6 for the precise characterization.

Finally, we define precisely what each assumption means in Table 1. For any space $\mathcal{A}$, let $\mathcal{M}(\mathcal{A})$ denote the set of all probability measures on $\mathcal{A}$.

**Neural Network:** $\quad \mathcal{F}_{\mathrm{NN}} \quad = \Big\{ f \in \mathcal{F} : \text{can be written as mat. mul. with scalar activations} \Big\}$

**NN with ERM:** $\quad \mathcal{F}_{\mathrm{ERM}} \quad = \Big\{ f \in \mathcal{F}_{\mathrm{NN}} : \exists \pi \in \mathcal{M}(\mathcal{I} \times [0,1]) \text{ s.t.}$

$$f = \underset{f' \in \mathcal{F}}{\arg\min} \underset{(X,Y)\sim\pi}{\mathbb{E}} \ell(f'(X), Y),$$

$\text{where } \ell \text{ is a Bregman loss function (see Banerjee et al., 2005)} \Big\}$

**Lipschitz:** $\quad \mathcal{F}_{\mathrm{Lip}}^{L} \quad = \Big\{ f \in \mathcal{F} : \underset{x,x'\in\mathcal{I}}{\sup} \dfrac{f(x) - f(x')}{\|x - x'\|_{\infty}} \le L \Big\}$

**Piecewise Affine:** $\quad \mathcal{F}_{\mathrm{PAff}} \quad = \Big\{ f \in \mathcal{F} : \text{can be written as affine on each piece of a partition of } \mathcal{I} \Big\}$

**Monotone:** $\quad \mathcal{F}_{\mathrm{Mono}} \quad = \Big\{ f \in \mathcal{F} : \forall x \le x', f(x) \le f(x') \Big\} \cup \Big\{ f : \forall x \le x', f(x) \ge f(x') \Big\}$ [3]

**Convex:** $\quad \mathcal{F}_{\mathrm{Convx}} \quad = \Big\{ f \in \mathcal{F} : \forall x, x' \in \mathcal{I} \; \forall \alpha \in [0,1],$

$$f(\alpha x + (1-\alpha)x') \le \alpha f(x) + (1-\alpha)f(x') \Big\}$$

**Affine:** $\quad \mathcal{F}_{\mathrm{Aff}}^{d} \quad = \Big\{ f \in \mathcal{F} : \exists A \in \mathbb{R}^{d} \; \exists b \in \mathbb{R} \text{ s.t. } \forall x \in \mathcal{I}, f(x) = A^{\mathrm{T}}x + b \Big\}$

**Constant:** $\quad \mathcal{F}_{\mathrm{Const}} \quad = \Big\{ f \in \mathcal{F} : \forall x, x' \in \mathcal{I}, f(x) = f(x') \Big\}.$

## B.3 ADDITIONAL IMPOSSIBILITY RESULTS

While Theorems 1 and 2 already provide impossibility for strong assumptions like monotonicity and piecewise affine, convexity is a particularly strong assumption since it restricts the output space of functions. In particular, no convex function can cross the diagonal line from $f_{\min}$ to $f_{\max}$, and hence it is possible that more information can be recovered from just these values. However, the next result shows that this information can only be used to possibly improve a constant $1/2$ to $1/4$, and arbitrary approximation is still impossible (unless the function is constant).

**Theorem 3.** *For all $D \in \mathcal{D}$ and $f \in \mathcal{F}_{\mathrm{Convx}}$, there exists $f' \in \mathcal{F}_{\mathrm{Convx}}$ such that $\Phi_{\min\max}(f) = \Phi_{\min\max}(f')$ and*

$$\left\| f' - D(\Phi_{\min\max}(f')) \right\|_{\infty} \ge \frac{f'_{\max} - f'_{\min}}{4}.$$

Similarly, a known Lipschitz constant implies local stability of $f$, which may be possible for a decoder to exploit. However, we show that this is also not possible in general (our result is stated in 1-dimension for simplicity, but could be extended trivially to arbitrary dimension using the sup norm definition of Lipschitz).

**Theorem 4.** *For all $D \in \mathcal{D}$, $L > 0$, and $f \in \mathcal{F}_{\mathrm{Lip}}^{L}$, there exists $f' \in \mathcal{F}_{\mathrm{Lip}}^{L}$ such that $\Phi_{\min\max}(f) = \Phi_{\min\max}(f')$ and if $2\,|f_{\max} - f_{\min}| \le L\,|x_{\max} - x_{\min}|$ then*

$$\left\| f' - D(\Phi_{\min\max}(f')) \right\|_{\infty} \ge \frac{f'_{\max} - f'_{\min}}{2}.$$

*Moreover, even if $2\,|f_{\max} - f_{\min}| > L\,|x_{\max} - x_{\min}|$,*

$$\left\| f' - D(\Phi_{\min\max}(f')) \right\|_{\infty} \ge L \max\Big\{ \min\{x_{\min}, x_{\max}\}, 1 - \max\{x_{\min}, x_{\max}\} \Big\}.$$

First, for all $f \in \mathcal{F}_{\mathrm{Lip}}^{L}$ it holds that $|f_{\max} - f_{\min}| \le L\,|x_{\max} - x_{\min}|$, so the first condition nearly captures all cases. As already argued, under this condition our lower bound is tight by Proposition 1. Even when the condition fails, our lower bound is zero if and only if $|x_{\max} - x_{\min}| = 1$

---

[3] For $d$-dimensional inputs, $x \le x'$ if and only if $x_j \le x'_j$ for all $j \in [d]$.

and $|f_{\max} - f_{\min}| > L/2$. This is nearly tight, since if $|f_{\max} - f_{\min}| = L$, then necessarily $|x_{\max} - x_{\min}| = 1$ and $f$ is linear and uniquely identifiable from $\Phi_{\min\max}(f)$ (and hence the lower bound must be zero in this case).

A similar condition can be used to provide a Lipschitz analogue of Theorem 2.

**Theorem 5.** *For all* $D \in \mathcal{D}$, $L > 0$, *and* $f \in \mathcal{F}_{\mathrm{Lip}}^L$ *such that* $2\,|f_{\max} - f_{\min}| \leq L\,|x_{\max} - x_{\min}|$, *there exists* $f' \in \mathcal{F}_{\mathrm{Lip}}^L$ *such that* $\Phi_{\min\max}(f) = \Phi_{\min\max}(f')$ *and*

$$\left\| \mathbf{1}_{f' > m_{f'}} - \mathbf{1}_{D(\Phi_{\min\max}(f')) > m_{f'}} \right\|_\infty \geq \mathbf{1}_{\sup_{x \in \mathcal{I}} f(x) \neq \inf_{x \in \mathcal{I}} f(x)}.$$

Finally, we have the following negative result for affine functions. Due to the extra structure imposed by an affine assumption, in more cases it is possible to fully recover $f$ from just the feature visualization. However, in the worst case, $f$ may still be completely unrecoverable. We show this for $d = 2$; a similar result can be shown in higher dimensions with more careful accounting of edge cases.

**Theorem 6.** *For all* $D \in \mathcal{D}$ *and* $f \in \mathcal{F}_{\mathrm{Aff}}^{d=2}$, *there exists* $f' \in \mathcal{F}_{\mathrm{Aff}}^{d=2}$ *such that* $\Phi_{\min\max}(f) = \Phi_{\min\max}(f')$ *and*

$$\left\| f' - D(\Phi_{\min\max}(f')) \right\|_\infty \geq \mathbf{1}_{x_{\min,1} \neq x_{\min,2}} \mathbf{1}_{x_{\max,1} \neq x_{\max,2}} \frac{f'_{\max} - f'_{\min}}{2}.$$

The same can also be shown for the analogue of Theorem 2.

**Theorem 7.** *For all* $D \in \mathcal{D}$ *and* $f \in \mathcal{F}_{\mathrm{Aff}}^{d=2}$, *there exists* $f' \in \mathcal{F}_{\mathrm{Aff}}^{d=2}$ *such that* $\Phi_{\min\max}(f) = \Phi_{\min\max}(f')$ *and*

$$\left\| \mathbf{1}_{f' > m_{f'}} - \mathbf{1}_{D(\Phi_{\min\max}(f')) > m_{f'}} \right\|_\infty \geq \mathbf{1}_{x_{\min,1} \neq x_{\min,2}} \mathbf{1}_{x_{\max,1} \neq x_{\max,2}}.$$

**Remark 2.** *Our negative results for affine functions rely on the constrained nature of the inputs. Without such constraints, the task of feature visualization would generally become even more difficult (and in practice, inputs are always bounded). However, specifically for affine functions, on unbounded inputs one could take advantage of the fact that the $\arg\max$ will be proportional to the weight vector, and hence could more reliably predict $f$ from $\Phi_{\min\max}(f)$. Similarly, adding regularization when computing $\Phi_{\min\max}$ could lead to reliably predicting $f$ even with bounded inputs.* ◁

### B.4 POSITIVE RESULTS

Finally, we state our positive result for very simple functions.

**Theorem 8.** *For all* $\mathcal{G} \in \{\mathcal{F}_{\mathrm{Aff}}^{d=1}, \mathcal{F}_{\mathrm{Const}}\}$ *there exists* $D \in \mathcal{D}$ *such that for all* $f \in \mathcal{G}$,

$$\left\| f - D(\Phi_{\min\max}(f)) \right\|_\infty = 0.$$

### B.5 PROOFS FOR SECTION 4

**Remark 3.** *Throughout, we prove impossibility results for 1-dimensional functions. The extension to multiple dimensions follows from using our constructions componentwise and then applying Lemma 3 or Lemma 4 as appropriate, which hold for any input dimension.* ◁

#### B.5.1 HELPER LEMMAS

To prove results for the first two columns on Table 1, we use the following lemma to characterize the performance of an arbitrary decoder $D \in \mathcal{D}$.

**Lemma 3.** *For any* $D \in \mathcal{D}$ *and* $f_1, f_2 \in \mathcal{F}$ *such that* $\Phi_{\min\max}(f_1) = \Phi_{\min\max}(f_2)$, *for some* $f \in \{f_1, f_2\}$

$$\left\| f - D(\Phi_{\min\max}(f)) \right\|_\infty \geq \frac{\|f_1 - f_2\|_\infty}{2}.$$

*Proof of Lemma 3.* Let $g = D(\Phi_{\min\max}(f_1)) = D(\Phi_{\min\max}(f_2))$ and let $x$ be such that $|f_1(x) - f_2(x)| = \|f_1 - f_2\|_\infty$. Then, since mean is less than max,

$$\frac{1}{2}|f_1(x) - f_2(x)| \leq \frac{1}{2}|f_1(x) - g(x)| + \frac{1}{2}|g(x) - f_2(x)| \leq \max_{f \in \{f_1, f_2\}} |f(x) - g(x)|.$$

$\square$

Then, for any $\mathcal{G} \subseteq \mathcal{F}$ of interest and any $f \in \mathcal{G}$, we simply have to find $f_1, f_2 \in \mathcal{G}$ such that $\Phi_{\min\max}(f) = \Phi_{\min\max}(f_1) = \Phi_{\min\max}(f_2)$ and $\|f_1 - f_2\|_\infty$ is appropriately large (where "large" will depend on $f$).

Similarly, we use the following lemma to prove results for the third column of Table 1.

**Lemma 4.** *For any $D \in \mathcal{D}$ and $f_1, f_2 \in \mathcal{F}$ such that $\Phi_{\min\max}(f_1) = \Phi_{\min\max}(f_2)$, for some $f \in \{f_1, f_2\}$*

$$\left\| \mathbf{1}_{f > m_f} - \mathbf{1}_{D(\Phi_{\min\max}(f)) > m_f} \right\|_\infty \geq \|\mathbf{1}_{f_1 > m} - \mathbf{1}_{f_2 > m}\|_\infty,$$

*where $m = m_{f_1} = m_{f_2}$.*

*Proof of Lemma 4.* Let $g = D(\Phi_{\min\max}(f_1)) = D(\Phi_{\min\max}(f_2))$ and let $x$ be such that $\left|\mathbf{1}_{f_1(x) > m} - \mathbf{1}_{f_2(x) > m}\right| = \|\mathbf{1}_{f_1 > m} - \mathbf{1}_{f_2 > m}\|_\infty$.

If $\|\mathbf{1}_{f_1 > m} - \mathbf{1}_{f_2 > m}\|_\infty = 0$ the result holds trivially, so suppose that $\|\mathbf{1}_{f_1 > m} - \mathbf{1}_{f_2 > m}\|_\infty = 1$. That is, $\mathbf{1}_{f_1(x) > m} \neq \mathbf{1}_{f_2(x) > m}$. If $\mathbf{1}_{g(x) > m} = \mathbf{1}_{f_1(x) > m}$, then

$$\left\| \mathbf{1}_{f_2 > m} - \mathbf{1}_{D(\Phi_{\min\max}(f_2)) > m} \right\|_\infty \geq \left|\mathbf{1}_{f_2(x) > m} - \mathbf{1}_{g(x) > m}\right| = 1.$$

Otherwise, if $\mathbf{1}_{g(x) > m} = \mathbf{1}_{f_2(x) > m}$, then

$$\left\| \mathbf{1}_{f_1 > m} - \mathbf{1}_{D(\Phi_{\min\max}(f_1)) > m} \right\|_\infty \geq \left|\mathbf{1}_{f_1(x) > m} - \mathbf{1}_{g(x) > m}\right| = 1.$$

That is,

$$\max_{f \in f_1, f_2} \left\| \mathbf{1}_{f > m} - \mathbf{1}_{D(\Phi_{\min\max}(f)) > m} \right\|_\infty \geq 1 = \|\mathbf{1}_{f_1 > m} - \mathbf{1}_{f_2 > m}\|_\infty.$$

$\square$

### B.5.2 PROOF OF PROPOSITION 1

Let $D(x_{\min}, x_{\max}, f_{\min}, f_{\max}) \equiv (1/2)(f_{\max} + f_{\min})$ and fix $f \in \mathcal{F}$. For any $x$,

$$f(x) - \frac{f_{\max} + f_{\min}}{2} \leq f_{\max} - \frac{f_{\max} + f_{\min}}{2} = \frac{f_{\max} - f_{\min}}{2}$$

and

$$\frac{f_{\max} + f_{\min}}{2} - f(x) \leq \frac{f_{\max} + f_{\min}}{2} - f_{\min} = \frac{f_{\max} - f_{\min}}{2}.$$

Thus,

$$\left| f(x) - \frac{f_{\max} + f_{\min}}{2} \right| \leq \frac{f_{\max} - f_{\min}}{2}.$$

Since $x$ was arbitrary, the result holds. $\square$

### B.5.3   PROOF OF THEOREM 1

First, suppose $0 \leq x_{\min} < x_{\max} \leq 1$.

Define

$$f_1(x) = \begin{cases} f_{\min} & x \in [0, (x_{\min} + x_{\max})/2] \\ \frac{2(f_{\max} - f_{\min})}{x_{\max} - x_{\min}} x + \frac{2f_{\min}x_{\max} - f_{\max}x_{\min} - f_{\max}x_{\max}}{x_{\max} - x_{\min}} & x \in [(x_{\min} + x_{\max})/2, x_{\max}] \\ f_{\max} & x \in [x_{\max}, 1] \end{cases}$$

and

$$f_2(x) = \begin{cases} f_{\min} & x \in [0, x_{\min}] \\ \frac{2(f_{\max} - f_{\min})}{x_{\max} - x_{\min}} x + \frac{f_{\min}x_{\max} + f_{\min}x_{\min} - 2f_{\max}x_{\min}}{x_{\max} - x_{\min}} & x \in [x_{\min}, (x_{\min} + x_{\max})/2] \\ f_{\max} & x \in [(x_{\min} + x_{\max})/2, 1]. \end{cases}$$

Since $\Phi_{\min\max}(f_1) = \Phi_{\min\max}(f_2) = \Phi_{\min\max}(f)$, and $f_1$ and $f_2$ are both monotone and piecewise affine, the result follows for $\mathcal{F}_{\mathrm{Mono}}$ and $\mathcal{F}_{\mathrm{PAff}}$ from applying Lemma 3 and observing that $\|f_1 - f_2\|_\infty \geq f_{\max} - f_{\min}$ (this occurs at $(x_{\min} + x_{\max})/2$).

If $0 \leq x_{\max} < x_{\min} \leq 1$, the same argument applies with $1 - f_1$ and $1 - f_2$.

Finally, when $x_{\min} = x_{\max}$, then $f_{\max} - f_{\min} = 0$ so the result holds trivially.

To prove the result for $\mathcal{F}_{\mathrm{NN}}$, note that we imposed no constraints on $f_{\max}$ or $f_{\min}$. Thus, for any $f \in \mathcal{F}_{\mathrm{NN}}$, we can construct $f_1$ and $f_2$. We then use that any piecewise affine function can be exactly represented by a sufficiently large neural network (Arora et al., 2018; Chen et al., 2022) to conclude $f_1, f_2 \in \mathcal{F}_{\mathrm{NN}}$.

The same argument applies to prove the result for $\mathcal{F}$, since clearly $f_1, f_2 \in \mathcal{F}$.

Finally, for $\mathcal{F}_{\mathrm{ERM}}$, we must construct appropriate distributions with conditional means $f_1$ and $f_2$ respectively (we already noted these are both elements of $\mathcal{F}_{\mathrm{NN}}$). For simplicity, define the joint distribution by $X \sim \mathrm{Unif}(\mathcal{I})$ and $Y|X \sim \mathrm{Ber}(f_j(X))$ for $j \in \{1, 2\}$. $\quad\square$

### B.5.4   PROOF OF THEOREM 2

We use $f_1$ and $f_2$ from Theorem 1, and recall that $m = (f_{\min} + f_{\max})/2$. Consider when $0 \leq x_{\min} < x_{\max} \leq 1$. Then, at $x = (x_{\min} + x_{\max})/2$, $f_1(x) = f_{\min} < m$ and $f_2(x) = f_{\max} > m$, so $\|\mathbf{1}_{f_1 > m} - \mathbf{1}_{f_2 > m}\|_\infty = 1$. The result then follows by Lemma 4. If $0 \leq x_{\max} < x_{\min} \leq 1$, the same argument applies with $1 - f_1$ and $1 - f_2$.

For $\mathcal{F}_{\mathrm{Convx}}$, we use $f_1$ and $f_2$ from the proof of Theorem 3 (Appendix B.5.5). Recall that $m = (f_{\min} + f_{\max})/2$. Consider when $x_{\max} = 1$ and $x_{\min} < 1$. Then, at $x = x_{\min}/4 + 3/4$, $f_1(x) = f_{\min}/4 + 3f_{\max}/4 > m$ and $f_2(x) = m$, so $\|\mathbf{1}_{f_1 > m} - \mathbf{1}_{f_2 > m}\|_\infty = 1$. The result then follows by Lemma 4. If $x_{\min} = 0$ and $x_{\max} > 0$, the same argument applies using $f_1'$ and $f_2'$ as defined in Appendix B.5.5.

If $x_{\min} = x_{\max}$ then $f_{\min} = f_{\max}$ and hence $f$ is constant, so the result holds trivially. $\quad\square$

### B.5.5   PROOF OF THEOREM 3

Note that for any $f \in \mathcal{F}_{\mathrm{Convx}}$, one of $x_{\min}$ or $x_{\max}$ are in $\{0, 1\}$.

First, consider $x_{\max} = 1$ and $x_{\min} < 1$. Define

$$f_1(x) = \begin{cases} f_{\min} & x \in [0, x_{\min}] \\ \frac{f_{\max} - f_{\min}}{1 - x_{\min}} x + \frac{f_{\min} - f_{\max}x_{\min}}{1 - x_{\min}} & x \in [x_{\min}, 1] \end{cases} \tag{10}$$

and

$$f_2(x) = \begin{cases} f_{\min} & x \in [0, (x_{\min} + 1)/2] \\ \frac{2(f_{\max} - f_{\min})}{1 - x_{\min}} x + \frac{2f_{\min} - f_{\max} x_{\min} - f_{\max}}{1 - x_{\min}} & x \in [(x_{\min} + 1)/2, 1]. \end{cases} \tag{11}$$

Clearly, $f_1, f_2 \in \mathcal{F}_{\text{Convx}}$ (since they are flat then linear with positive slope) and $\|f_1 - f_2\|_\infty \geq (f_{\max} - f_{\min})/2$ (this occurs at $x = (x_{\min} + 1)/2$). Since $x_{\min} = 0$ implies that $x_{\max} = 1$ by convexity, this case also covers $x_{\min} = 0$.

Second, consider $x_{\max} = 0$ and $x_{\min} > 0$. Using Eqs. (10) and (11), define $g_1(x) = f_1(1 - x)$ and $g_2(x) = f_2(1 - x)$. These also satisfy $g_1, g_2 \in \mathcal{F}_{\text{Convx}}$ (since they are linear with negative slope then flat) and $\|g_1 - g_2\|_\infty \geq (f_{\max} - f_{\min})/2$ (this occurs at $x = x_{\min}/2$). Since $x_{\min} = 1$ implies that $x_{\max} = 0$ by convexity, this case also covers $x_{\min} = 1$.

Finally, if $x_{\min} = x_{\max}$ then $f_{\min} = f_{\max}$ and the result holds trivially. $\qquad\square$

### B.5.6    PROOF OF THEOREM 4

When $2|f_{\max} - f_{\min}| \leq L|x_{\max} - x_{\min}|$, the proof of Theorem 1 applies since $f_1, f_2 \in \mathcal{F}_{\text{Lip}}^L$.

Otherwise, suppose that $0 \leq x_{\min} < x_{\max} \leq 1$. Define

$$f_1(x) = \begin{cases} f_{\min} & x \in [0, x_{\min}] \\ \frac{f_{\max} - f_{\min}}{x_{\max} - x_{\min}} x + \frac{f_{\min} x_{\max} - f_{\max} x_{\min}}{x_{\max} - x_{\min}} & x \in [x_{\min}, x_{\max}] \\ f_{\max} & x \in [x_{\max}, 1] \end{cases}$$

and

$$f_2(x) = \begin{cases} -L(x - x_{\min}) + f_{\min} & x \in [0, x_{\min}] \\ \frac{f_{\max} - f_{\min}}{x_{\max} - x_{\min}} x + \frac{f_{\min} x_{\max} - f_{\max} x_{\min}}{x_{\max} - x_{\min}} & x \in [x_{\min}, x_{\max}] \\ -L(x - x_{\max}) + f_{\max} & x \in [x_{\max}, 1]. \end{cases}$$

Recall that by definition of $f \in \mathcal{F}_{\text{Lip}}^L$, $|f_{\max} - f_{\min}| \leq L|x_{\max} - x_{\min}|$, so $f_1, f_2 \in \mathcal{F}_{\text{Lip}}^L$.

If $x_{\min} > 1 - x_{\max}$, then $\|f_1 - f_2\|_\infty \geq Lx_{\min}$ (which is realized at $x = 0$), and otherwise $\|f_1 - f_2\|_\infty \geq L(1 - x_{\max})$ (which is realized at $x = 1$).

If $0 \leq x_{\max} < x_{\min} \leq 1$, the same argument applies with $1 - f_1$ and $1 - f_2$. $\qquad\square$

### B.5.7    PROOF OF THEOREM 5

Since $2|f_{\max} - f_{\min}| \leq L|x_{\max} - x_{\min}|$, the proof of Theorem 2 applies because $f_1, f_2 \in \mathcal{F}_{\text{Lip}}^L$. $\square$

### B.5.8    PROOF OF THEOREM 6

When $d = 2$, any $f \in \mathcal{F}_{\text{Aff}}^{d=2}$ satisfies $f(x) = ax_1 + bx_2 + c$ for some $a, b, c \in \mathbb{R}$. Given $\Phi_{\text{min max}}(f) = (x_{\min}, f_{\min}, x_{\max}, f_{\max})$, the compatible $f \in \mathcal{F}_{\text{Aff}}^{d=2}$ are those $f_c$ such that

$$a = \frac{x_{\max,2} f_{\min} - x_{\min,2} f_{\max} + (x_{\min,2} - x_{\max,2})c}{x_{\min,1} x_{\max,2} - x_{\max,1} x_{\min,2}}$$

$$b = \frac{-x_{\max,1} f_{\min} + x_{\min,1} f_{\max} + (x_{\max,1} - x_{\min,1})c}{x_{\min,1} x_{\max,2} - x_{\max,1} x_{\min,2}},$$

where $c$ is a free parameter (with the only constraint that $f_c(x) \in [0, 1]$ for all $x \in \mathcal{I}$).

Since $f$ is affine, $x_{\min}$ and $x_{\max}$ must both occur at one of the four corners of $[0, 1]^2$. Note that $a$ and $b$ above are undefined for some of these combinations, which we now enumerate.

If $x_{\min} = x_{\max}$, then $f$ is constant and can be recovered exactly. If $x_{\min} = (0,0)$, then necessarily $c = f_{\min}$, so $f$ can be recovered exactly. Similarly, if $x_{\max} = (0,0)$ then necessarily $c = f_{\max}$.

Moreover, there are other cases where $f$ can be recovered. If $x_{\min} = (1,1)$ and $x_{\max} \in \{(1,0),(0,1)\}$, then one of $a$ or $b$ do not depend on $c$ and hence $c$ can be directly recovered. The same is true when $x_{\max} = (1,1)$.

There are two possibilities left.

1) $x_{\min} = (0,1)$ and $x_{\max} = (1,0)$:
$$a = f_{\max} - c$$
$$b = f_{\min} - c.$$

Take $c_1 = f_{\min}$ and $c_2 = f_{\max}$ to get
$$f_1(x) = (f_{\max} - f_{\min})x_1 + f_{\min}$$
and
$$f_1(x) = (f_{\min} - f_{\max})x_2 + f_{\max}.$$
These still have $\Phi_{\min\max}(f) = \Phi_{\min\max}(f_1) = \Phi_{\min\max}(f_2)$, but $\|f_1 - f_2\|_\infty \geq f_{\max} - f_{\min}$ (this is realized at $x = (1,1)$).

2) $x_{\min} = (1,0)$ and $x_{\max} = (0,1)$:
$$a = f_{\min} - c$$
$$b = f_{\max} - c.$$

Take $c_1 = f_{\min}$ and $c_2 = f_{\max}$ to get
$$f_1(x) = (f_{\max} - f_{\min})x_2 + f_{\min}$$
and
$$f_1(x) = (f_{\min} - f_{\max})x_1 + f_{\max}.$$
These still have $\Phi_{\min\max}(f) = \Phi_{\min\max}(f_1) = \Phi_{\min\max}(f_2)$, but $\|f_1 - f_2\|_\infty \geq f_{\max} - f_{\min}$ (this is again realized at $x = (1,1)$). The result holds by then applying Lemma 3. $\square$

### B.5.9    PROOF OF THEOREM 7

This follows directly from applying Lemma 4 to the functions constructed in the proof of Theorem 6 (Appendix B.5.8). $\square$

### B.5.10    PROOF OF THEOREM 8

First suppose that $f \in \mathcal{F}_{\text{Aff}}^{d=1}$. That is, there exists $a, b$ such that $f(x) = ax + b$ for all $x$. Given $\Phi_{\min\max}(f)$, define
$$a_f = \frac{f_{\max} - f_{\min}}{x_{\max} - x_{\min}}$$
and
$$b_f = \frac{x_{\max}f_{\min} - x_{\min}f_{\max}}{x_{\max} - x_{\min}}.$$
Set $D(\Phi_{\min\max}(f)) = [x \mapsto a_f x + b_f]$. Since there is a unique affine function passing through both $(x_{\min}, f_{\min})$ and $(x_{\max}, f_{\max})$, and $D(\Phi_{\min\max}(f))$ is an affine function passing through both of these, this implies that $D(\Phi_{\min\max}(f)) \equiv f$.

If $f \in \mathcal{F}_{\text{Const}}$, then there exists $y \in [0,1]$ such that $f_{\min} = f_{\max} = y$ and $f(x) = y$ for all $x$. Define $D(\Phi_{\min\max}(f)) \equiv f_{\min}$, which implies that $D(\Phi_{\min\max}(f)) \equiv f$. $\square$

### B.6 CAN WE MOVE BEYOND WORST-CASE ANALYSES?

The theoretical results above are of a worst-case nature (for every claim, we construct a counterexample to refute the claim). Of course, another interesting question to ask is: How likely are these counterexamples to appear in reality when we generate the function $f$ by training a neural net using SGD? Here, we briefly discuss our results in the context of this average-case perspective.

On the one hand, a strength of our counterexamples is that they can be realized by very simple functions, and hence we are not "cherry-picking" convoluted functions that SGD will not learn. Similarly, our counterexamples can be easily extended to a family of functions that are rich in the space of all functions in the model class, again in contrast to having only a single, unrealistic function that works as a counterexample. On the other hand, it is very possible that while these counterexamples are easily learned by SGD, they are more or less likely depending on, say, the specific training data. Unfortunately, to answer this question, it seems one would have to have a rather refined understanding of the distribution of models learned from training on realistic data via SGD, which would resolve some very large open problems in learning theory along the way. We pose it as an open problem to extend our approach of proving negative results for feature visualization by incorporating the learning algorithm.

### B.7 RELATIONSHIP OF THEORETICAL RESULTS TO PSYCHOPHYSICAL EXPERIMENTS

Borowski et al. (2021) and Zimmermann et al. (2021) performed psychophysical experiments to investigate the tness of feature visualizations for human observers. Both papers find that natural highly activating images are more interpretable (as measured by human prediction performance) compared to feature visualizations. A candidate explanation for this behaviour is our analysis in Section 3, showing that for last-layer Inception-V1 feature visualizations, those visualizations are processed along very different paths for most of the network (compared to natural images as a baseline).

The task used by Borowski et al. (2021) is related to the third column of Table 1. They asked participants to predict which one of two natural images is strongly activating for a certain unit based on maximally and minimally activating feature visualizations for that unit. This can be seen as an easier version of the task in Table 1: Borowski et al. (2021) did not use random test samples but instead two curated samples, out of which one has extremely high and one has extremely low activations. They find that that humans are able to do this task above chance.

At first glance, this result seems to contradict our theoretical finding from Theorem 2, which states that reliable prediction is impossible unless additional assumptions about the function are known. However, there is no contradiction: our theory allows for the possibility of a specific function (e.g., a specific neural network unit) and a specific decoder (e.g., a human observer) to be aligned in the sense that predictions about the function can happen to be correct—however, for every function $f$ for which the decoder is correct there is a function $f'$ from the same function family for which the decoder is wrong (in spirit, a case of "no free lunch" for feature visualization). If an observer gets significantly above chance in one case, they would pay the price by being significantly below chance in the other case. To the best of our knowledge, there is currently no way for the observer to know in advance whether they're visualizing a function $f$ for which their decoding is aligned or a function $f'$ for which their decoding leads to the wrong conclusions. It is an interesting open question to develop a practical and rigorous approach to distinguish these cases, perhaps relying on additional information such as the data distribution (e.g., does ImageNet lead to benign $f$ more often?) and the training procedure (e.g., does SGD lead to benign $f$ more often?).

## C METHOD DETAILS

### C.1 CLASSIFIER TRAINING (SECTION 2.1)

For classifier training, we create a dataset by combining $1,281,167$ images from the training set of the ImageNet 2012 dataset and $472,500$ synthetic images. These synthetic images are the (intermediate) results of the feature visualization optimization process. Specifically, for $1,000$ classification units in the last layer of an ImageNet-trained InceptionV1 network, we run the optimization process with the parameters used by (Olah et al., 2017) 35 times each, resulting in $35,000$ unique optimiza-

tion trajectories. We logarithmically sample $15$ (intermediate) steps from the optimization trajectory, resulting in $525,000$ synthetic images in total. Finally, we split the synthetic images into $472,500$ ($= 90\%$) training and $52,500$ ($= 10\%$) testing images. Note that we use different units for the two sets. We train a model implementing the simple six layer CNN architecture displayed in Table 3 for $8$ epochs on the aforementioned dataset with an SGD optimizer using a learning rate of $0.01$, momentum of $0.9$ and weight decay of $0.00005$. The classifier achieves a near-perfect accuracy of $99.49\%$ on the held-out test set ($99.66\%$ and $99.31\%$ for natural input and feature visualizations, respectively).

| Type | Size/Channels | Activation | Stride |
|---|---|---|---|
| Conv $3 \times 3$ | 16 | ReLU | 3 |
| Conv $5 \times 5$ | 16 | ReLU | 2 |
| Conv $5 \times 5$ | 16 | ReLU | 2 |
| Conv $5 \times 5$ | 16 | ReLU | 2 |
| Conv $5 \times 5$ | 16 | ReLU | 2 |
| Conv $3 \times 3$ | 16 | ReLU | 2 |
| Flatten | - | - | - |
| Linear | 1 | - | - |

Table 3: Architecture of the classifier used to detect feature visualizations.

### C.2 FEATURE VISUALIZATION FIGURES (SECTION 2)

Throughout the paper, feature visualizations were generated using the `lucent` library (Greentfrapp, v0.1.8), version v0.1.8. Per default, we used `thresholds=(512,)` except for Figure 3 where the five images at different points in the optimization trajectory are shown (specifically, `thresholds=(1,8,32,128,512)`). For Figure 1, we used the thresholds that visually looked best (in line with existing literature: there is no principled way to determine the threshold); specifically `thresholds=(512,512,512,6,32,6)` for the six visualizations from left to right (for the three rightmost images, higher thresholds produced qualitatively similar yet oversaturated images). In terms of transformations during feature visualization, `transforms=lucent.optvis.transform.standard_transforms + [center_crop(224, 224)]` was used. The image was parameterized via `param_f=lambda: lucent.optvis.param.image(224, batch=1)`.

For Figure 1, a natural image was embedded into the weights of a single convolutional layer, `torch.nn.Conv2d`, with `kernel_size=224, stride=1, padding=0, dilation=1, groups=1, bias=True, padding_mode='zeros'`). To this end, the image was loaded and the layer weights were set to the corresponding image values, divided by $224^2$ to avoid a potential overflow. Architecturally, the layer received the standard image input and its ouptut was appended to the output of the desired layer (e.g., `softmax2_pre_activation_matmul`) where it was used in the role of $D$ from Figure 4.

### C.3 SANITY CHECK (SECTION 3)

**Motivation: relationship between path similarity and Spearman correlation.** In Section 3, we describe that different processing paths lead to different activation similarity as measured through Spearman correlation. We here attempt to explain this relationship in a bit more detail. For the context of our analysis, we define a path as a (sub-)graph of a directed acyclic graph (DAG, a neural network or sub-network in our case), starting at the input nodes (first layer units) and ending at a single unit (the unit for which the analysis is performed). How can we quantify the overlap between two different paths, layer by layer? If a node is in the subgraph forming the path, the node is assigned a value of 1; if it is not, it is assigned a value of 0. Then, layer-wise overlap can be quantified by the Spearman correlation, which is exactly zero if there is only chance overlap, exactly 1.0 if there is perfect overlap, and exactly -1.0 if the units in a certain layer (corresponding to two different paths) are perfectly anticorrelated. Similarly, in the non-binary case (such as a path formed by activation patterns for natural images vs. feature visualization images, which is what we consider for the similarity analysis), the values assigned to the node are simply the activations, and the same

analysis can be applied. Since we only care about the path similarity and not about whether this similarity is a linear relationship, Spearman's rank-order correlation is the correct measure to use here (while the Pearson correlation, plotted in Figure 10 for comparison, is a measure of a linear relationship).

**Methods.** Performing a full comparison is computationally expensive: Inception-V1's largest layer (`conv2d0_pre_relu_conv`) contains 802,816 values; computing the Spearman correlation for this takes about one third of a second. Inception-V1 has 138 layers and sub-layers (see Figure 7 x-axis labels for a list). Even if we just consider the comparison between natural images vs. natural images of a different class, for the ImageNet-1K validation set this amounts to 138 (= number of layers) $\cdot$ 50 $\cdot$ 50 (= number of comparisons between two specific classes of the ImageNet validation split) $\cdot \frac{1000 \cdot 1001}{2} - 1000$ (= number of comparisons when comparing each class with each other class except for itself) $= 172,327,500,000$ comparisons. With 3 comparisons per second, this amounts to about *eighteen hundred years* required to do the full comparison. In order to make this more feasible, we chose the following approach. We randomly selected 10 classes via `numpy.random.seed(42);` `randomly_selected_class_indices = sorted(numpy.random.choice(1000,` `10))`, resulting in `randomly_selected_class_indices=[20, 71, 102, 106,` `121, 270, 435, 614, 700, 860]` and obtained 10 feature visualizations per class. Images that were not correctly classified by the model (wrong top-1 classification) were excluded from the comparison since those images do not constitute natural images for which the corresponding unit is selective for.

From this point onward, when computing the Spearman and Pearson correlations, we only performed every 10th comparison for natural images vs. natural images of the same class; every 100th comparison for natural images vs. natural images of a different class, and every 5th comparison for natural images vs. feature visualizations of the same class. For Cosine similarity (which is much faster), we performed every single comparison.

**Raw and normalized plots.** For each metric, *absolute* values are plotted in Figures 7, 9, and 11. For Figures 6, 8, and 10, *normalized* values are plotted. For these plots, we normalized the data according to the raw absolute values, i.e., such that natural images vs. natural images of the same class is set to 1.0 and natural images vs. natural images of a different class is set to 0.0 since it makes sense to interpret similarity results relative to those two extreme baselines. A tiny number of layers was excluded from the normalized comparison if the green and black points from the absolute plots differed by strictly less than a threshold of 0.01. To reduce noise in the orange curve (natural images vs. feature visualizations of the same class), we smoothed the curve by convolving it with `scipy.ndimage.convolve` using a window size of 7 and the following uniform weights: `np.ones(windowsize)/windowsize`. The shaded blue area corresponds to the standard deviation of the orange data points, convolved over a window of of size `std_windowsize=5` which is then (for the lower bound of the blue area) subtracted from the orange curve, and (for the upper bound of the blue area) added to the orange curve; thus in total the blue area area vertically extends two standard deviations. The idea behind this is to give a rough visual estimate of the standard deviation range that the orange values have at certain points throughout the network. By itself, it does not provide an indication of statistical significance.

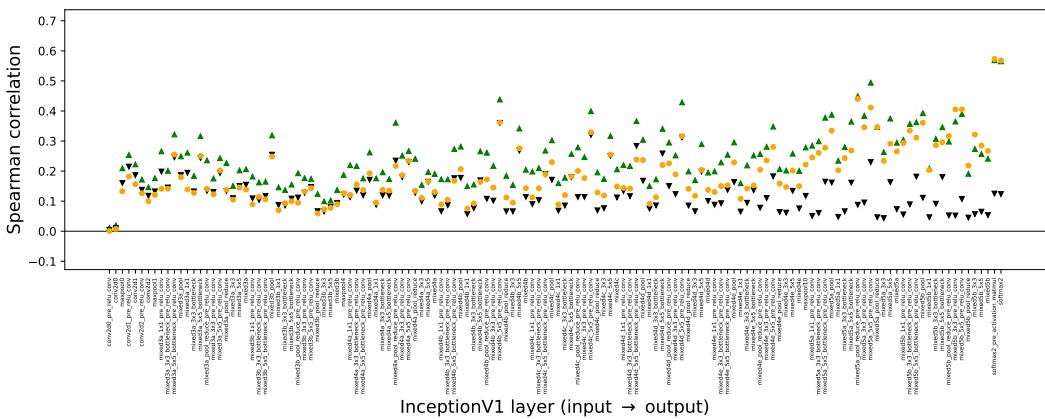

Figure 7: Absolute similarity (Spearman).

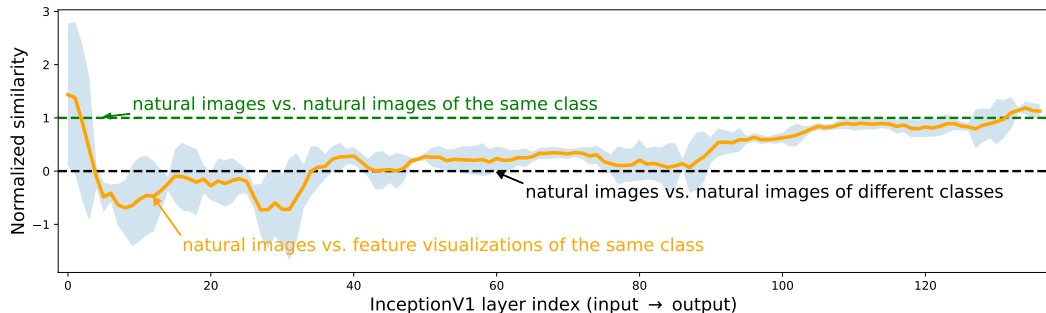

Figure 8: Normalized similarity (Cosine).

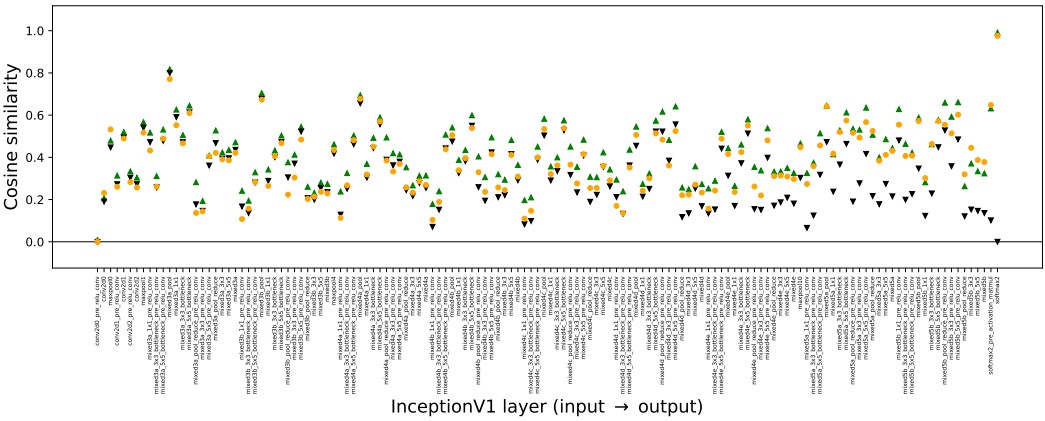

Figure 9: Absolute similarity (Cosine).

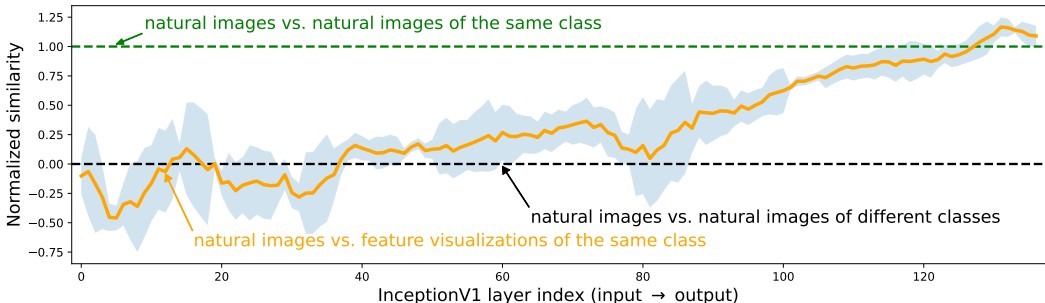

Figure 10: Normalized similarity (Pearson).

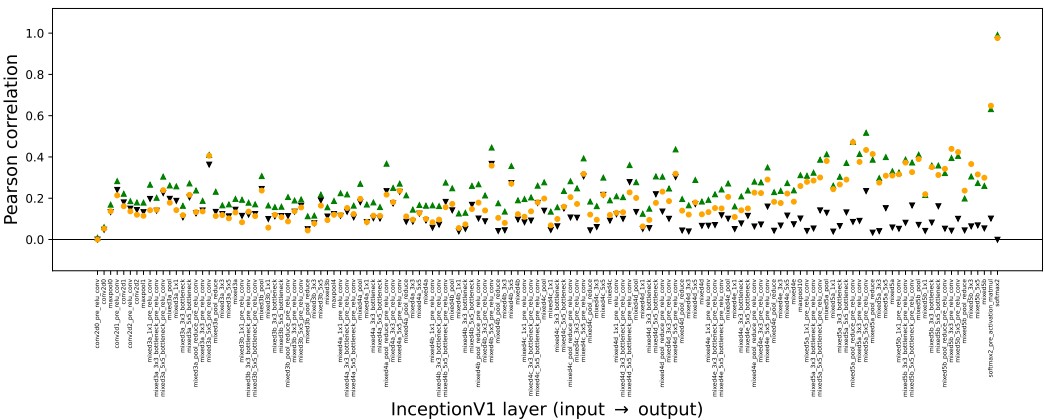

Figure 11: Absolute similarity (Pearson).

## C.4 LINEARITY EXPERIMENTS (SECTION 5)

As our theory (see Table 1) predicts that the simpler the function class of a network's unit is the easier it may be to interpret through feature visualizations, we set out to measure one type functional simplicity. Specifically, we approximate how *linear* a function's optimization trajectory is by computing how aligned its gradients are. The hypothesis is that more linear units (or visualization paths) might be more interpretable.

To measure the interpretability of a unit we use the experimental data provided by Zimmermann et al. (2023). Based on the experimental paradigm by Zimmermann et al. (2021), Zimmermann et al. (2023) tested how well humans can differentiate maximally and minimally activating images for individual units of a CNN when supported with explanations in the form of feature visualizations (see Appendix B.7 for details). We use the experimental data of $84$ units as well as the $M = 20$ maximally activating natural dataset samples (from ImageNet) they provided.

**Quantifying degree of nonlinearity through gradient angles** To measure the linearity of a unit we compute the following quantity for each unit: We start from a maximally activating dataset sample $x_i^s$ and perform feature visualization to iteratively optimize this image to further increase the unit's activation. We use standard hyperparameters and optimize for $N = 512$ steps. During optimization we record the normalized gradients with respect to the current image $(\hat{g}_j(x_i^s))_{j=1,\ldots,N}$ and compute the angle between successive steps:

$$\forall j = 1, \ldots, N-1: \quad a_j(x_i^s) := \measuredangle(\hat{g}_j(x_i), \hat{g}_{j+1}(x_s^s)). \tag{12}$$

We take the average over all $M$ maximally activating images and denote the *average gradient path angle* as:

$$\forall j = 1, \ldots, N-1: \quad \mathrm{AGPA}_j := \frac{1}{M} \sum_{i=1}^{M} a_j(x_i^s). \tag{13}$$

Finally, we denote the average of the average gradient path angle AGPA as the average gradient angle:

$$\mathrm{AGA} = \frac{1}{N-1} \sum_{i=1}^{N-1} \mathrm{AGPA}_i. \tag{14}$$

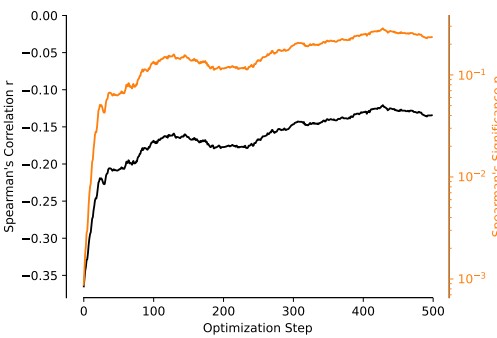 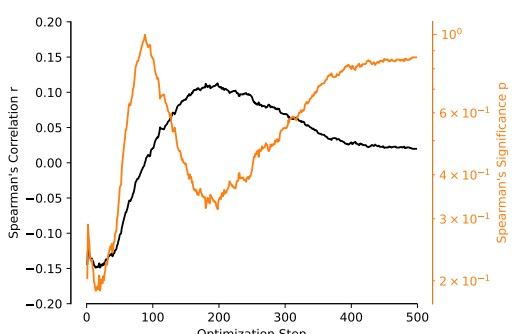

(a) Average gradient path angle $\mathrm{AGPA}_k$. Strikingly, only the first few gradient angles are strongly and significantly correlated with the units' interpretability.

(b) Average line distance path $\mathrm{AGLD}_k$. Interestingly, while we see a anti-correlation at the beginning (cf. Figure 12a), this turns into a weak correlation. However, these correlations are not significant.

Figure 12: Development of Spearman's rank correlation between the units' interpretability score and the (a) average gradient path angle $\mathrm{AGPA}_k$ and (b) average line distance path $\mathrm{ALDP}_k$ as a function of the number of optimization steps $k$ to consider.

To answer our initial question—whether simpler/more linear units are more interpretable—we now measured the rank correlation between the average gradient angle and the interpretability scores by Zimmermann et al. (2023) based on the paradigm of Zimmermann et al. (2021). Intriguingly, we find a significant correlation (Spearman's $r = -.36$, $p = .001$) between this human interpretability score and our path linearity measure (lower angle means higher interpretability) for the beginning of the trajectory (e.g., $n = 2$). Linearity at later steps in the trajectory does not seem to contribute much to human interpretability, thus increasing $n$ to include all 512 steps decreases the overall correlation. The results depending on path length are plotted in Figure 12a.

**Quantifying degree of nonlinearity through deviations from linear interpolation** There are many different ways that could be used to measure path or unit linearity. As a more global measure, we also tested another one: Here, we begin by computing a linear interpolation between the maximally activation data samples we initialize the optimization with $x_i^s$ and the final visualization $x_i^f$:

$$\{ z \mid x_i^s + \alpha(x_i^f - x_i^s) \quad \forall \alpha \in [0,1] \}. \tag{15}$$

Next, for each step $j$ of the optimization process we compute the distance of the current image $x_j(x_i^s)$ to to the linear interpolation

$$d_j(x_i^s) = d\left( x_j((x_i^s)), \{ z \mid x_i^s + \alpha(x_i^f - x_i^s) \quad \forall \alpha \in [0,1] \} \right), \tag{16}$$

where $d(\cdot, \cdot)$ represents the $\ell_2$ distance. Analogously to the computation above, we then take the mean over the different start images and define this property as the average line distance path

$$\forall j = 1, \ldots, N-1: \quad \mathrm{ALDP}_j := \frac{1}{M} \sum_{i=1}^{M} \frac{d_j(x_i^s)}{d(x_i^2, x_i^f)}, \tag{17}$$

feature visualization trajectory          natural ImageNet validation images

Figure 13: **Natural vs. synthetic distribution shift for a robust model.** As shown above for the Inception-V1 model in Figure 3, there is still a clear distribution shift between feature visualizations (left) and natural images (right) an adversarial robust model (ResNet50). Visualizations at different steps in the optimization process for a randomly selected unit in the last layer of standard, unmanipulated but robust ResNet50; randomly selected ImageNet validation samples (excluding images containing faces).

where we normalize the distances from the line with the distance between start and end point of the optimization trajectory to make these values scaleless. Finally, by averaging again over optimization steps final average line distance:

$$\text{ALD} = \frac{1}{N-1} \sum_{i=1}^{N-1} \text{ALDP}_i \,. \tag{18}$$

The lower the ALD value, the smaller is the deviation of the optimization path from a line. For this global measure, we see a non-significant relation between the average line distance and the interpretability scores (Spearman's $r = 0.02$, $p = 0.86$). Analogously to the local measure explained above (gradient angle), we zoom into these results again in Figure 12b. While there might be a weak anti-correlation at the beginning of the optimization path (which later turns into a weak correlation), none of these are significant.

**Interpretation.** We believe there is more to be understood: while it is intriguing that linearity at the beginning of the optimization trajectory is predictive of a human interpretability score, the global measure of linearity (through distance to linear interpolation) does not seem to be predictive and more investigations are needed to fully understand this phenomenon—as we say in the main paper, this can only be considered "very preliminary evidence". A promising way forward could be to enforce (or optimize for) different properties in neural networks and measure whether this improves human interpretability.

## D  FOOLING FEATURE VISUALIZATIONS OF ADVERSARIALLY ROBUST MODELS

In Section 2, we presented two proofs of concept that feature visualizations can manipulated: one based on a fooling circuit, and one leveraging silent units. We empirically demonstrated this for two standard vision models. Our theoretical results in Table 1 in Section 4 indicate that feature visualizations become more reliable for networks with a sufficiently small Lipschitz constant. At the same time, a lower Lipschitz constant is also connected with higher adversarial robustness of models (Hein & Andriushchenko, 2017). Therefore, a natural question is: Can feature visualizations of (more) robust models still be manipulated? We investigate this question for adversarially trained ResNet50 ($\ell_\infty, \epsilon = 4/255$) by Salman et al. (2020).

**Fooling circuit** Integrating a fooling circuit is guaranteed to manipulate the visualizations of a model without changing its overall behavior as long as one can clearly distinguish between feature visualizations and standard dataset examples (see Section 2.1). Therefore, to demonstrate the this approach also works for an adversarial robust model, it suffices to show that we can almost surely detect whether an input is a feature visualization. Qualitatively, this is suggested by an analogy of Figure 3, i.e., a visual comparison of feature visualizations and dataset samples, in Figure 13. Quantitatively, we tested the *same* feature visualization detector that we used for a non-robust Inception-V1 model in Section 2.1 for the robust ResNet50 and see strong generalization results: the binary classifier's accuracy is still very high at 99.19 %.

**Silent units** To apply the methodology presented in Section 2.2 to an adversarial robust model, only a single change needs to be implemented: Namely, we noticed that the ranges of activations

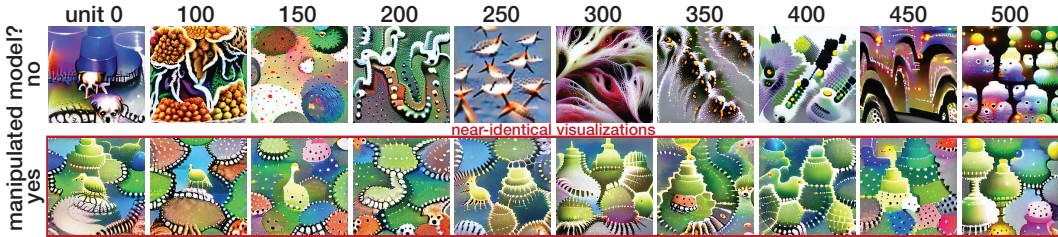

Figure 14: **Leveraging silent units to produce identical visualizations throughout a layer of a robust model.** Replication of Figure 5 for a robust ResNet50 model. The top row shows feature visualizations for units of a layer (block 4-1, conv 2) in a robust but unmanipulated ResNet-50. For the bottom row, we manipulate the model such that the feature visualizations of all units become near-identical (indicated by the red box).

caused by feature visualizations and dataset examples, respectively, are closer for a robust than for a non-robust model. For some units, there even exist natural dataset examples that elicit slightly higher activation than feature visualizations. We, therefore, need to adjust how we choose the bias $b$ in Eq. (11): Instead of using a value proportional to the maximal activation value recorded on any natural test sample, we use a value proportional to the 99th percentile of the activation range for test samples. While this change ensures that we can manipulate the visualizations of more/all units, it comes with a small price: Namely, the network's behavior on natural samples does not remain unchanged (as for a non-robust ResNet50, cf., Section 2.2) but drops very slightly from 63.924 % to 63.784 %. The analogue of Figure 5 for the robust model, Figure 14, shows that even a robust model can be manipulated to show near-identical feature visualizations.

# E  DO FEATURE VISUALIZATIONS FAIL THE SANITY CHECK SIMPLY BECAUSE THEY ARE OUT-OF-DISTRIBUTION?

In Section 3, we showed that feature visualizations fail a sanity check. An anonymous reviewer asked whether this might be due to the case that they are out-of-distribution (compared to natural images)—an interesting question that we decided to investigate. We therefore performed the analysis for two complementary additional datasets: ImageNet-V2 (Recht et al., 2019) which has the same classes as original ImageNet and only a small distribution shift as evidenced by a roughly 10–15% accuracy drop of standard models, and NINCO (Bitterwolf et al., 2023), a dedicated out-of-distribution detection dataset that specifically contains no classes that are part of original ImageNet. We run the sanity check analysis with those datasets separately by computing the same baselines as before (natural images vs. natural images of the same class; natural images vs. natural images of different classes) with the difference that those natural images are now sampled from the respective OOD dataset, not from ImageNet. We then plot the previously computed similarity between feature visualizations vs. natural ImageNet images of the same class in relationship to those new baselines. The key idea is that if a simple out-of-distribution shift is responsible for strongly decreased processing similarity, then those new baselines should be drastically lower. At the same time, the normalized similarity between feature visualizations and natural ImageNet images (plotted in orange) should be a lot higher than it is in Figure 6 since it is now normalized with respect to the out-of-distribution baselines.

As we can see for ImageNet-V2 in Figure 15 and for NINCO in Figure 17, this is *not* the case: even though there is a substantial distribution shift, the similarity between same-class images of those OOD datasets is still a lot higher than the natural-vs-feature-visualization similarity on ImageNet throughout the network (except for the last few layers). We conclude from this analysis that simple distribution shifts are insufficient to explain the different processing paths of feature visualizations— either the distribution shift does not play a role, or the distribution shift from feature visualizations is much larger even than the ImageNet-NINCO distribution shift (keeping in mind that NINCO is a dedicated out-of-distribution detection dataset aimed at providing a much more systematic shift than many other datasets).

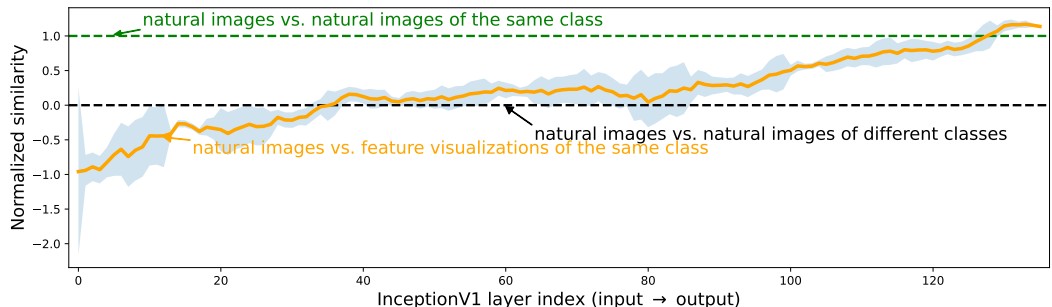

Figure 15: Normalized similarity (Spearman) for ImageNet-V2.

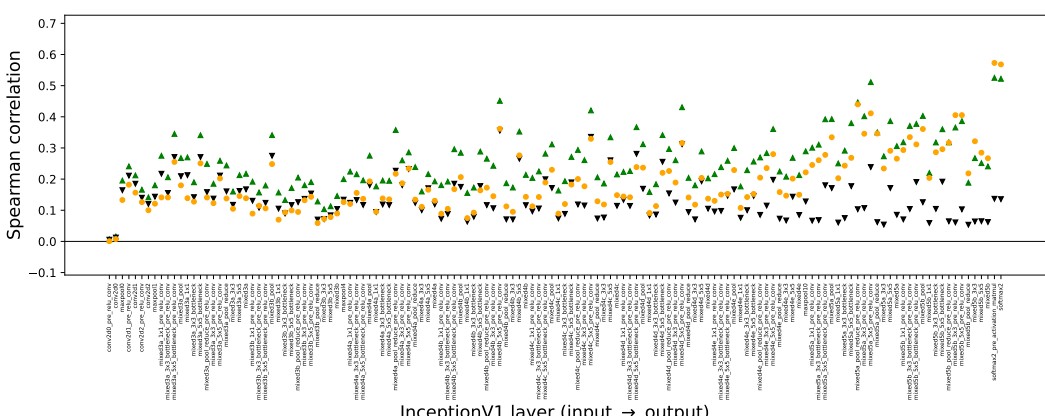

Figure 16: Absolute similarity (Spearman) for ImageNet-V2.

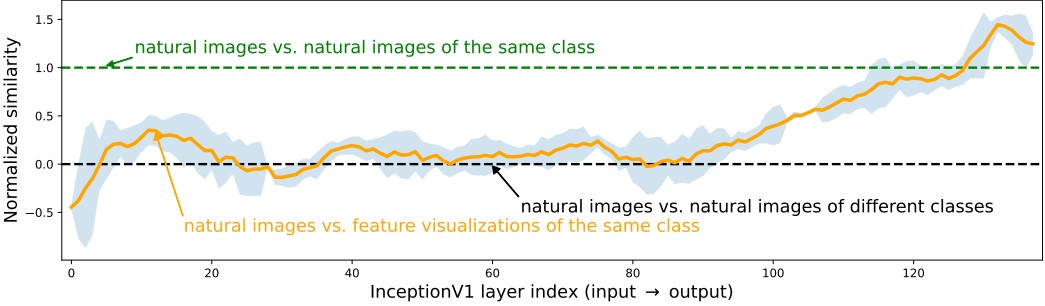

Figure 17: Normalized similarity (Spearman) for NINCO.

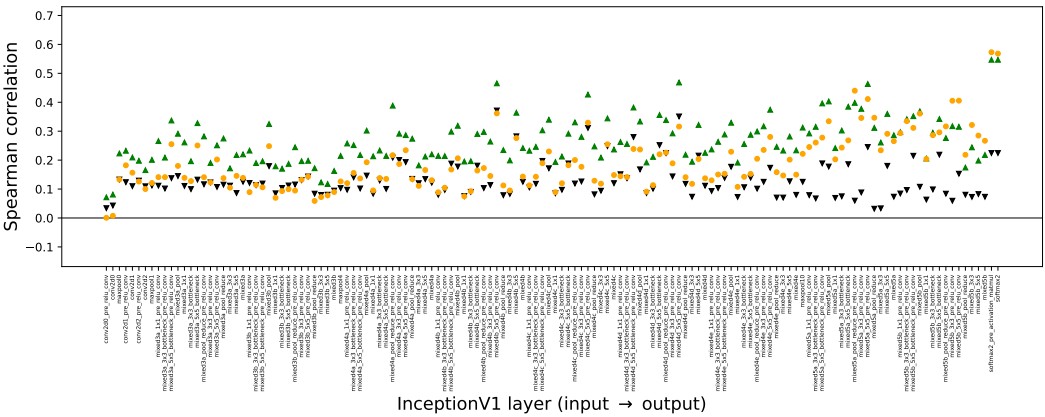

Figure 18: Absolute similarity (Spearman) for NINCO.

## F    BROADER IMPACTS

Our paper investigates the reliability of feature visualizations. Overall, we expect this to contribute to better scrutiny towards existing interpretability methods, which hopefully inspires the development of more reliable interpretability methods in the future, as well the development of models that incorporate certain reliably "interpretability-enabling" assumptions right from the start, rather than being faced with the (sometimes impossible) task of post-hoc interpretability through feature visualizations.

In terms of potential negative impact, the fooling methods developed here could be used to deceive an entity (e.g., a model auditor or regulator) as described in Section 2 and Appendix A.1. That being said, we believe that the risk is lower if this knowledge is public—it would be much more problematic to believe that feature visualizations can be taken at face value, because then whoever designs a fooling circuit would be met with an unsuspecting audience.

## G    LIMITATIONS

We see the following potential limitations:

1. We design methods that fool feature visualizations. Once it is known that a certain fooling method might be used, it is easy to develop a detection mechanism. That said, the space of potential fooling methods is vast. Therefore, developing a specific detection mechanism would probably lead to a pattern similar to adversarial attacks and defenses: after an attack is developed, a detection/circumvention method defends, which is then again circumvented by a revised attack/fooling method.

2. The fooling methods that we developed in Section 2 assume bad intent. Most models are developed with good intent. However, we believe that the reliability of interpretability methods should not rely on assuming good intentions. The experiments in Section 3 and the theory from Section 4 are independent of good/bad intent assumptions.

3. No sanity check is perfect, and like any sanity check, ours is just a necessary but not a sufficient condition for a reliable feature visualization. For instance, if the training data contains spurious correlations (all cows are on grass); a unit in the network is selective for grass and not for cows but the feature visualization shows realistic-looking cows on grass (rather than just grass), then the visualizations would pass the sanity check without raising a warning about the spurious correlation present in the visualization. We would love to see more research on sanity checks—to the best of our knowledge we provide the first one which hopefully serves as a motivation for research on both better visualizations and more sanity checks.

4. The potential assumptions on the function space listed in Table 1 are not exhaustive. It is possible that other assumptions enable stronger prediction. Furthermore, our theory is a

worst-case analysis. It may be possible to go beyond worst-case analyses—an aspect we discuss in Appendix B.6—but the strength of our theoretical counterexamples is that they can be realized by very simple functions, hence we are not "cherry-picking" complicated functions that SGD would never learn.

5. The investigated networks, Inception-V1 and ResNet-50, are of course not exhaustive either. None of our methods is specific to those networks. This means that other networks could be equipped with a fooling circuit, too. At the same time, the empirical results might look different for other networks, which would be an interesting direction to explore in future work.

6. One could argue that the fooling circuit in Figure 4 only deceives a user when looking at unit $A$, whereas the other units still have their original visualization. That's correct: the fooling circuit manipulates the visualization of $A$ but not of e.g., units $D$ or $F$. From a single unit perspective, this is already problematic since it means we can't trust a unit's visualization. It would be interesting avenue for future work to develop networks where every single unit's visualization is misleading.

7. There is no one definition of what it means to "understand" or "explain" a neural network, since those are very vague terms. We seek to be precise about our definition and motivate it with expectations about feature visualization stated in the literature (Appendix A.2), but we realize that this means not everyone's notion of "understanding" / "explaining" neural networks can be captured by our definition.

## H    IMAGE SOURCES

**Figure 1.** "Girl with a Pearl Earring" by Johannes Vermeer was downloaded from here and is public domain according to the website. "Puppies" (West Highland White Terrier puppies) by Lucie Tylová, Westik.cz was downloaded from here and is licensed under CC BY-SA 3.0 according to the website. "Mona Lisa" by Leonardo da Vinci was downloaded from here and is public domain according to the website. All three images were cropped to size $224 \times 224$ pixels. The photographs were then embedded in the weights of a single convolutional layer and to some degree recovered by the feature visualization method, subject to distortion by the method's transformations.

