# OpenReview forum: "Don't trust your eyes: on the (un)reliability of feature visualizations"
_ICLR.cc/2024/Conference — Submitted to ICLR 2024_

### Official Review · Reviewer_o92S · 2023-10-30

**Soundness:** 3 good
**Presentation:** 4 excellent
**Contribution:** 4 excellent
**Rating:** 8
**Confidence:** 3

**Summary:**

This paper questions the reliability of feature visualizations, a common tool used to understand individual features/neurons of a network. They present three lines of evidence against trusting feature visualizations:
1. Adversarial - by developing a new "adversarial attack" to fool feature visualizations
2. Empirical - by showing feature visualizations are processed differently than natural images by the network
3. Theoretical - by showing we can't know much about a neuron based on feature visualization alone, even under strong assumptions

**Strengths:**

Clearly written, good motivation and discussion of related work. Highlights an important problem, and the two attack strategies presented is both simple and effective.

**Weaknesses:**

I am not fully convinced by section 3 and the path perspective, I think it is plausible a feature visualization could be informative and useful even if it looks different from natural images, and a visualization of late layer features does not need to be processed similarly to natural images at least on early layers.

**Questions:**

While this paper only focuses on feature visualization, it seems like these issues reflect a bigger issue with using a low number of highly activating examples to describe a unit. Do you think this work should also raise concerns against using let's say 5 most highly activating natural images to describe a neuron, as this is a common alternative to using feature visualizations?

---

> ### Author Response · Authors · 2023-11-16
> **Author response**
>
> Dear reviewer o92S,
>
> Thank you for your review, we’re glad to hear that you found the paper **“clearly written”**, **“highlighting an important problem”** with **“excellent presentation & contribution”**.
>
> **Point 1** *“a visualization of late layer features does not need to be processed similarly to natural images at least on early layers”*:
>
> We fully agree that differences in early layers wouldn’t be problematic. As an example, it would be totally fine for an “airplane” visualization to have different color statistics or low-level features compared to natural airplane images. It would only be problematic if the “airplane” visualization lacks mid- and high-level features that belong to airplanes. Our Figure 6 didn’t really allow a reader to assess which layers correspond to low-, mid-, and high-level features. In the light of your comment, we have therefore updated Figure 6 with annotations that show not only the name of the layer (previously we used just an uninformative index) but also an annotation copied from https://distill.pub/2017/feature-visualization/appendix/ that describes layers as responding to “textures”, “object parts”, “sophisticated concepts” etc.
> Given this annotation, it becomes clear that feature visualizations are processed differently from natural images not only in low-level layers but also in layers corresponding to object parts, complex and sophisticated concepts - precisely the kinds of features that the visualizations were designed to ‘explain’ and visualize. If a feature visualization wants to succeed in explaining how natural input is processed, it needs to better capture those features. We have added the following to Section 3: *“While it would be fine for a feature visualization to show different low-level features compared to natural images, any visualization that seeks to explain how natural input is processed should capture similarities in mid- and high-level layers that are described by Olah et al. (2017) as responding to ‘object parts’ and ‘complex/sophisticated concepts’.”*
>
> **Point 2** *“Do you think this work should also raise concerns against using let's say 5 most highly activating natural images to describe a neuron”:*
>
> That’s an interesting question! Our paper covers three perspectives and we thus have three answers to this question.
>
> From the *adversarial perspective*, we could easily use our method from Section 2.2 to build a network where the top k natural images don’t correspond to what the unit is usually selective for; this would simply mean setting the bias parameter b to a lower value such that it would only suppress activations up to, say, the 95th percentile of natural input. A recent paper specifically looked into manipulating the top-k activating images; we cited it in the related work section as “A complementary approach is proposed by Nanfack et al. (2023), which changes highly activating dataset samples” (https://arxiv.org/pdf/2306.07397.pdf).
>
> From the *empirical perspective*, highly activating natural images would pass the sanity check since highly activating natural images are, by definition, natural images that highly activate a unit, and they would thus be processed like other natural images.
>
> From the *theoretical perspective*, our impossibility results can be extended to many variations on using the argmax for feature visualization, including using the most activating dataset samples as explanations. Essentially, as long as the feature visualization method does not narrow down the function space too much, our results will apply. It is easy to see that two simple (e.g., piecewise linear) functions could have the same 5 (or 10, etc.) local (arg)maxima and yet behave very differently even quite near these local maxima, and hence our theorems could be extended.
>
> Thus, in summary, highly activating natural images would pass our sanity check, but it would still be possible to construct networks that show misleading highly activating natural images, which is a case that’s covered by the theory. Since this is a question we feel might interest readers as well, we have added a dedicated and detailed explanation to Appendix A.3 (“Relationship to highly activating natural samples”), essentially covering the points we describe above. Thanks again for the question!

---

> > ### Comment · Reviewer_o92S · 2023-11-23
> > **Response**
> >
> > Thanks for the interesting response! This mostly addresses my remaining questions and I retain my rating of accept.

---

### Official Review · Reviewer_MuGq · 2023-10-31

**Soundness:** 3 good
**Presentation:** 2 fair
**Contribution:** 2 fair
**Rating:** 6
**Confidence:** 3

**Summary:**

This manuscript studies issues of treating feature visualization as a means to explain the behavior of vision classification models. Their main contributions are 2 types of ways to generate different feature visualizations and show deep models essentially treat natural images and feature visualizations differently. The rest is a set of theoretical analysis showing what feature visualization could do.

**Strengths:**

This work has an overall good presentation of results. It designs two counterfactual examples to show the unreliability of feature visualizations and any downstream understandings based on feature visualization are also broken. Authors nicely convey the messages behind the theories and highlight the main take-ways.

**Weaknesses:**

Lack of clear definitions on threat models. I am not entirely sure how the fool circuit works. It is mostly I don't know what the authors assume an adversary could do and couldn’t do. Moreover, I don’t see definitions of the unreliability of feature visualizations. Everything seems to be hand-wavey in the first empirical sections.

Theories have weak connections with experiments. Although I understand the theorems are just proofs of concepts and demonstrate the boundary of understanding from feature visualizations, they lack strong connections with what you experimented in the previous sections. Are those experiments motivated by theoretical analysis? Is any take-away from the theory empirically validated by the experiments? Why suddenly the theory bounds the function using Linf norms? When I read the theories I mainly feel these results imply what could be done in the future work, as the linearity discussion mentioned by authors. However, they lack a strong connection with your empirical results shown above.

**Questions:**

Have you experimented with adversarially robust models? A large body of work has shown robust models have better saliency explanations, better representations and thus better geometry of the output. Thus, how does the quality of the model affect the quality/reliability of feature explanations? But, before answering the last question, please define the reliability first.

---

> ### Author Response · Authors · 2023-11-17
> **Author response (1/2)**
>
> Dear reviewer MuGq,
>
> Thank you for your review and questions. Please find answers to your questions below:
>
> 1. *Lack of clear definitions on threat models [...] It is mostly I don't know what the authors assume an adversary could do and couldn’t do.*
>
> That’s a great point, we can be more explicit about the threat model in Section 2. The “adversarial perspective” (Section 2) assumes that a model can be arbitrarily manipulated (e.g. by changing connectivity patterns or slightly modifying the architecture) while the interpretability technique (feature visualization) is fixed and no aspect of this technique (hyperparameter, random starting point) can be changed by a person with malicious intent. This is different to the setting in adversarial robustness, where the model is fixed and the input is manipulated. We have revised the beginning of Section 2 to include the following explanation on the threat model:
>
> “... since feature visualizations have been proposed as model auditing tools (Brundage 2020) that should be integrated “into the testbeds for AI applications” (Nguyen 2019), it is important to understand whether an adversary (i.e., someone with malicious intent) might be able to construct a model such that feature visualizations are manipulated. This corresponds to a threat scenario where the model itself can be arbitrarily changed while the interpretability technique (feature visualization) is kept fixed without control over hyperparameters or the random starting point. For example, a company may be interested in hiding certain aspects of its model's behavior from a third-party (e.g. regulator) audit that uses feature visualizations. In this context, our demonstration of unreliability relates to a line of work on deceiving other interpretability methods (described in detail in Appendix A.1). We do not know whether such a scenario could become realistic in the future, but as we stated above the main motivation for this section is to provide a proof of concept by showing that one can build networks that deceive feature visualizations.”
>
> 2. *Definition of unreliability*
>
> We argue that a visualization method is unreliable if it is possible to modify a network such that (1) its overall behavior stays the same but (2) its visualizations are changed (into arbitrary patterns). Since this definition wasn’t mentioned clearly in the text before, we have now added the following definition to Section 2 to define unreliability:
> “One important requirement for interpretability is that the explanations are reliable. We use the following definition of unreliability: A visualization method is unreliable if one can change the visualizations of a unit without changing the unit’s behavior on (relevant) test data. More formally, this can be expressed as: Let $\mathcal{U}$ denote the space of all  units. A visualization method $m$ is unreliable if $\exists u,v \in \mathcal{U}: m(u) = m(v) \land \neg u \overset{bhv}{\sim} v$, where $\overset{bhv}{\sim}$ denotes an  equivalence class of equal behavior.”
>
> 3. *Theories have weak connections with experiments.*
>
> Thank you for highlighting that we can increase the connection between theory and experiments. In response to your comment, we have revised the text to make the two main connections we see more clear:
>
> First, the theory of Section B.1 is directly related to the experiments of Section 2.1, proving that the fooling circuit will always work the way we observe in the experiments. We have added the following text to the end of Section 2.1 in the revision: “In Section B.1, we formalize this fooling circuit, and prove that it will always behave in the way we observe in our experiments.”
>
> Second, the main role of the impossibility theorems is to establish that many informal claims in the literature about the performance of feature visualization cannot be true in general. The connection to the empirical results here is to show that regardless of what fooling method is used, there is no defense that will provide general guarantees for feature visualization without very strong assumptions on the network. We have added the following text to the start of Section 4 in the revision: “A natural question arising from our experiments is whether the limitations of feature visualization that we have shown experimentally are always present, or if they can be avoided in certain situations. Towards addressing this, we now ask: When are feature visualizations guaranteed to be reliable, i.e. guaranteed to produce results that can be relied upon?”

---

> ### Author Response · Authors · 2023-11-17
> **Author response (2/2)**
>
> 4. *Why does the theory bound the function using $L_{\infty}$ norms?*
>
> This is a great question. To provide clarity for all readers, we have added the following text to the beginning of Section 4.1 in the revision: “Throughout, we measure the accuracy of predicting f using the $L_{\infty}$ norm. This is primarily for convenience; the equivalence of $L_p$ norms on bounded, finite-dimensional spaces implies we could prove the same results with $L_p$ for any $p$ at the expense of dimension-dependent constants. This captures many cases of interest: $L_p$ for $p \in \{1,2\}$ is the standard measure of accuracy for regression, and for $f$ that outputs bounded probabilities, the logistic loss is equivalent to $L_2$ up to constants.”
>
> 5. *Have you experimented with adversarially robust models?*
>
> That’s a fascinating suggestion! There’s a connection between adversarial training and our theory. Our proofs (as summarized in Table 1) predict that feature visualizations cannot be guaranteed to be reliable even for L-Lipschitz networks, unless we use a weak notion of ‘prediction’ in combination with a sufficiently low Lipschitz constant. As increasing the adversarial robustness of models is related to decreasing their Lipschitz constant, one might wonder whether feature visualizations become more reliable for adversarially robust models. In response to your question, we therefore stress-tested both of our fooling methods from Section 2 (fooling circuit & silent units) for an adversarially robust model trained by Salman et al. (2020). The results are very clear: both methods succeed in fooling the visualizations of adversarially robust models out of the box. The detailed results for those two experiments can be found in Appendix D. Thanks again for your suggestion!
>
> (In case you’re interested in the details: The method in Section 2.1 works as long as the feature visualizations of a network have a sufficiently large perceptual difference to natural dataset samples - still the case for adversarially robust models. In fact, we can use the existing classifier trained to distinguish natural samples vs. feature visualizations for a non-robust model out of the box without any re-training; this classifier achieves an accuracy of 99.19% for visualizations from an adversarially robust model. The method in Section 2.2 leverages the difference in activation levels caused by feature visualizations and dataset samples. While this activation gap appears to be a bit smaller, it still exists and can be used to fool feature visualizations on adversarially robust models with impressive results as shown in Figure 14.)
>
> Thanks again for your comments and questions, and kindly let us know if we have been able to address your suggestions and concerns!

---

> > ### Author Response · Authors · 2023-11-22
> > **Any concerns remaining?**
> >
> > Dear reviewer MuGq,
> >
> > Given that the discussion period is closing soon, and since you're the only reviewer whose score is currently leaning towards rejection, we were wondering whether there are any remaining concerns from your side that we might be able to address?

---

> > > ### Comment · Reviewer_MuGq · 2023-11-23
> > > **Thanks for the response**
> > >
> > > The rebuttal has addressed most of my initial concerns. As a result, I decide to increase the score, i.e. 5 -> 6.

---

### Official Review · Reviewer_CUhk · 2023-11-10

**Soundness:** 3 good
**Presentation:** 3 good
**Contribution:** 3 good
**Rating:** 6
**Confidence:** 3

**Summary:**

This paper proposes a fancy method to challenge the reliability of feature visualization. The authors therefore support three aspects to fool，sanity-check and proof based the assumed theories. They show us how sensitive the feature visualization is when applying just a few units on the neural network which is a novel way to interpret the back-box of NN. And the authors indeed provided general and detailed proofs in the mian paper to illustrate their proposed method.

**Strengths:**

1. The proposed method in this paper is kind of interesting and novel，which inspires us to challenge the current feature visualization methods and bring with a new point.

2. This paper is well written.

3. The mathematical proofs are plentiful and interesting.

**Weaknesses:**

1. This paper should provide us with more quantitive results on some popular benchmarks to better demonstration the fooling or sanity-checking of intermediate feature can change the model‘s recognition ability.

2. The NN model interpretation method for dense prediction tasks，like class activation map is widely used in weakly learning community. It will be more convincing if the author can provide some visualization results using their proposed method to see how will the final target object saliency map changes.

**Questions:**

Please refer to the weakness part.

---

> ### Author Response · Authors · 2023-11-14
> **Author response & follow-up questions**
>
> Dear reviewer CUhk,
>
> Thanks for taking the time to review our paper. We’re happy to hear that you found our paper **“well written”**, with **“interesting & plentiful proofs”** and an **“interesting & novel method”**.
>
> We wanted to reply quickly since we have a few follow-up questions (see below).
>
>
> **Point 1** *"This paper should provide us with more quantitive results on some popular benchmarks to better demonstration the fooling or sanity-checking of intermediate feature can change the model‘s recognition ability."*:
>
> You’re right: many robustness methods seek to “fool” a network in the sense that the network’s accuracy should drop given manipulated input. However, the goal of the fooling circuit is to NOT change the model’s recognition ability. Instead, we fool an interpretability method: we want to maintain the network’s object recognition ability (quantified by its classification accuracy) while fooling the feature visualization. This is validated by showing that the resulting network maintains identical ImageNet-1K validation accuracy (as described in Section 2) while fooling visualizations (as demonstrated in Figures 1, 2 and 5). Does this explanation make sense? Furthermore, would your point be addressed by providing more quantitative results showing that the network’s accuracy on standard (e.g., classification) benchmarks is *not* changed/affected by our fooling method (if so, we’re happy to test the model on more benchmarks)?
>
> **Point 2** *"The NN model interpretation method for dense prediction tasks, like class activation map is widely used in weakly learning community. It will be more convincing if the author can provide some visualization results using their proposed method to see how will the final target object saliency map changes."*:
>
> We’re not sure we fully understand the question. Saliency maps ask “which pixels provide evidence for a specific class given this particular image”. In contrast, feature visualizations ask “how does the maximally activating input for a unit look like” (without explaining any specific input image). Both areas are interesting but the methods are different. In particular, in this paper we focus on feature visualizations and provide ample samples (Figures 1,2,3,5). Our methods neither aim nor attempt to fool saliency maps; this has already been done extensively by prior work (e.g. Adebayo et al. 2018, Nie et al. 2018). Could you let us know whether this clarifies the matter, or elaborate if it doesn’t?

---

### Official Review · Reviewer_NDh1 · 2023-11-10

**Soundness:** 4 excellent
**Presentation:** 3 good
**Contribution:** 3 good
**Rating:** 8
**Confidence:** 4

**Summary:**

Feature visualization has so far been a central tool to generate insights about the features encoded by the units of computer vision systems. This paper asks: how reliable are feature visualizations? They first present a proof of concept that feature visualizations can be fooled as they show that they can make a model behave differently when classifying natural images vs. when generating feature visualizations. Motivated by this proof of concept they propose, a sanity check to make sure that feature visualizations generated by a method are indeed processed by a model in the same way as the natural images used for its training, as well as, a theoretical framework regarding the reliability of feature visualizations. Finally, the theoretical framework points out toward simpler models as more likely to be interpretable with the help of feature visualization and they offer preliminary evidence backing up that hypothesis.

**Strengths:**

- The paper is well-written and clear
- They highlight limitations of feature visualization that are necessary to keep in mind (especially when used to audit black-box models)
- The sanity check is motivated, sound, and a valuable contribution to the community

**Weaknesses:**

- I would advise for softening some of the claims/contextualization of the problem/results of the paper:
	- While feature visualization is one of the major types of interpretability methods in computer vision, it seems misleading to talk about "widespread use", as it is still a relatively minor method compared to truly widespread attribution methods (as feature visualization already suffers from other more practical limitations).
	-  While the proofs of concept are informative and something to keep in mind, they are extreme cases unless one is using the method for auditing external models (which is far from the common scenario). The scope of the proof of concept might be clear in the conclusion but not as much in the rest of the paper.

**Questions:**

- Have the authors experimented with robust classifiers? What exactly causes feature visualization to be processed differently from natural inputs is a very interesting question.
 - No sanity check is perfect, especially if it becomes a metric to optimize for. Do the authors have intuition about the limitation of the sanity check, i.e., how could the sanity check be fooled?
- While the experiments proposed are indeed not model-specific, half seem to be done using an Inception and the other half a ResNet50, is there a reason for this inconsistency?
- Regarding the sanity check I would be very curious to see the results from other baseline to further explore the causes that could explain the difference.
	- (a) Is it simply a question of the ood-ness of the feature visualization as pointed out by Figure 3? In that case, I would be curious to see as extra baselines such as natural images vs natural images of the same class from another dataset (e.g., [1]) and natural images vs natural images of different classes from another dataset.
	- (b) Is it a question of the naturalness of the feature visualization as they look quite different from natural images or simply a problem of the fact that the visualization is optimized? In that case, I would be curious to see the results from methods that enforce more natural-looking images either by making use of a generative model ([2-3]) or that simply use specific regularization for that ([4]).
- Not putting this in weakness, but I am not sure what to make of the theoretical section. Even though feature visualizations have been shown, at times ([4-5-6-7]), to offer insights about the network's unit activation, the main takeaway of this section, for now, is to caution that there is no guarantee that this is a general trend or that the insights are not only at a coarse level, which is important to keep in mind but it does not invalidate the empirical results previously cited. That being said, if one of the conclusions of this section is that alternatives to the current paradigm of activation maximization should be seriously considered by the community, it would be insightful if the authors could suggest possible alternatives that would overcome some of the limitations pointed out in this work.
- With regard to softening the claims (and w.r.t the previous point), one concrete suggestion is, in the conclusion: ": Given that our theory proves that visualizing black-box systems via feature visualizations based on activation maximization (i.e., the current dominant paradigm) is impossible -> "Given that our theory proves that there is no guarantee that ... is possible"


[1] Bitterwolf, Julian, Maximilian Müller, and Matthias Hein. "In or Out? Fixing ImageNet Out-of-Distribution Detection Evaluation." arXiv preprint arXiv:2306.00826 (2023).

[2] Anh Nguyen, Alexey Dosovitskiy, Jason Yosinski, Thomas Brox, and Jeff Clune. Synthesizing the preferred inputs for neurons in neural networks via deep generator networks. Advances in neural information processing systems, 29, 2016.

[3] Anh Nguyen, Jeff Clune, Yoshua Bengio, Alexey Dosovitskiy, and Jason Yosinski. Plug & play generative networks: Conditional iterative generation of images in latent space. In Proceedings of the IEEE conference on computer vision and pattern recognition, pages 4467–4477, 2017.

[4] Fel, Thomas, et al. "Unlocking Feature Visualization for Deeper Networks with MAgnitude Constrained Optimization." arXiv preprint arXiv:2306.06805 (2023).

[5] Judy Borowski, Roland Simon Zimmermann, Judith Schepers, Robert Geirhos, Thomas SA Wallis, Matthias Bethge, and Wieland Brendel. Exemplary natural images explain CNN activations better than state-of-the-art feature visualization. In International Conference on Learning Representations, 2021

[6] Roland S Zimmermann, Judy Borowski, Robert Geirhos, Matthias Bethge, Thomas Wallis, and Wieland Brendel. How well do feature visualizations support causal understanding of CNN activations? Advances in Neural Information Processing Systems, 34:11730–11744, 2021.

[7] Roland S Zimmermann, Thomas Klein, and Wieland Brendel. Scale alone does not improve mechanistic interpretability in vision models. arXiv preprint arXiv:2307.05471, 2023

---

> ### Author Response · Authors · 2023-11-17
> **Author response (1/2)**
>
> Dear reviewer NDh1,
>
> Thanks for your very helpful review, it’s rare to see such a detailed review and we very much appreciate it! We’re happy to hear that you found the soundness “**excellent**” and the paper “**well-written and clear**” with contributions that are “**valuable to the community**”. Please find a point-by-point response below.
>
> 1. *Contextualizing / describing “widespread use” of feature visualizations:*
>
> We agree that contextualization is important. It is correct that, given the size of the overall ML community, feature visualizations are not a generally widespread tool. What we meant to say is that feature visualizations are one of the standard tools within the mechanistic interpretability community (which we think resonates with your review stating that feature visualizations are a “central tool to generate insights about the features encoded by the units of computer vision systems”). We identified a number of sentences where we stated “widely used / widespread use”  and contextualized to make this more clear. For instance, we now write “Despite its widespread use within the mechanistic interpretability community” / “Feature visualizations based on activation maximization are a widely used tool within the mechanistic interpretability community”.
>
> 2. *Contextualizing / The scope of the proof of concept might be clear in the conclusion but not as much in the rest of the paper.*
>
> It seems that the right way to contextualize the proof of concept is to emphasize that it requires access to the model. We have thus revised the introduction summary bullet point 1. to state “Thus, feature visualizations can be deceived if one has access to the model” and the conclusion of Section 2 which now reads “This establishes that feature visualizations can be arbitrarily manipulated if one has access to the model” to make this more clear.
>
> 3. *Have the authors experimented with robust classifiers?*
>
> That’s a fascinating suggestion! There’s a connection between adversarial training and our theory. Our proofs (as summarized in Table 1) predict that feature visualizations cannot be guaranteed to be reliable even for L-Lipschitz networks, unless we use a weak notion of ‘prediction’ in combination with a sufficiently low Lipschitz constant. As increasing the adversarial robustness of models is related to decreasing their Lipschitz constant, one might wonder whether feature visualizations become more reliable for adversarially robust models. In response to your question, we therefore stress-tested both of our fooling methods from Section 2 (fooling circuit & silent units) for an adversarially robust model trained by Salman et al. (2020). The results are very clear: both methods succeed in fooling the visualizations of adversarially robust models out of the box. The detailed results for those two experiments can be found in Appendix D. Thanks again for your suggestion!
>
> (In case you’re interested in the details: The method in Section 2.1 works as long as the feature visualizations of a network have a sufficiently large perceptual difference to natural dataset samples - still the case for adversarially robust models. In fact, we can use the existing classifier trained to distinguish natural samples vs. feature visualizations for a non-robust model out of the box without any re-training; this classifier achieves an accuracy of 99.19% for visualizations from an adversarially robust model. The method in Section 2.2 leverages the difference in activation levels caused by feature visualizations and dataset samples. While this activation gap appears to be a bit smaller, it still exists and can be used to fool feature visualizations on adversarially robust models with impressive results as shown in Figure 14.)
>
> 4. *No sanity check is perfect, especially if it becomes a metric to optimize for. Do the authors have intuition about the limitation of the sanity check, i.e., how could the sanity check be fooled?*
>
> That’s an interesting point to think about! We’ve added the following example of a limitation to the limitations section (App. G):
> “No sanity check is perfect, and like any sanity check, ours is just a necessary but not a sufficient condition for a reliable feature visualization. For instance, if the training data contains spurious correlations (all cows are on grass); a unit in the network is selective for grass and not for cows but the feature visualization shows realistic-looking cows on grass (rather than just grass), then the visualizations would pass the sanity check without raising a warning about the spurious correlation present in the visualization. We would love to see more research on sanity checks - to the best of our knowledge we provide the first one which hopefully serves as a motivation for research on both better visualizations and more sanity checks.”

---

> ### Author Response · Authors · 2023-11-17
> **Author response (2/2)**
>
> 5. *While the experiments proposed are indeed not model-specific, half seem to be done using an Inception and the other half a ResNet50, is there a reason for this inconsistency?*
>
> Yes! We were worried that readers and reviewers might think the experiments/methods are somewhat model-specific or do not generalize across architectures if we just use a single network. We therefore made sure to include two different model families. We realize that’s a bit confusing without explanation and have thus revised the intro to Section 2.2 to include the following: “We designed this method to show that fooling can easily be achieved in different ways and across different architectures. In order to demonstrate that different model families can be fooled, we here consider a different architecture (ResNet-50 instead of Inception-V1) and a randomly selected intermediate layer instead of the last layer (but note that the approach is not specific to a certain architecture or layer).”
>
> 6. *Regarding the sanity check I would be very curious to see the results from other baseline to further explore the causes that could explain the difference.*
>
> We’re working on this and will update the response once we have results. [EDIT: please see https://openreview.net/forum?id=OZWHYyfPwY&noteId=H0vbRC6wgf below for our update and Appendix E for the results]
>
> 7. *Theory doesn’t invalidate the empirical results previously cited [4-5-6-7].*
>
> Correct, our theory does not invalidate the empirical results of previous studies [4-7]. Instead, it complements and completes their findings: It is true that previous quantitative studies [5-7] demonstrated that feature visualizations do indeed provide (some) information for understanding units of practically relevant neural networks. However, we’d like to point out an important caveat: although these papers used a (relatively) weak notion of “understanding”, neither feature visualizations nor maximally activating dataset samples yield (close to) perfect understanding - there is still a gap that can be closed. This gap becomes substantially larger when making the notion of understanding just slightly harder [7]. We argue that this might suggest there is more going on inside the networks that these explanations do not explain - potentially because the explanation methods implicitly put strong assumptions on the networks that are, in practice, only weakly or coincidentally fulfilled. Our appendix B.7 includes a discussion on the relationship between our findings and previous psychophysical empirical studies.
>
> 8. *It would be insightful if the authors could suggest possible alternatives that would overcome some of the limitations.*
>
> We agree with you that finding a new paradigm that overcomes the reported issues will be an exciting result. While we would have loved to provide a more reliable feature visualization method for general black-box neural networks, our theory proves that this cannot be guaranteed to be reliable: irrespective of the nuances of different feature visualization methods based on activation maximization, much more knowledge about the ‘black-box’ is necessary to make reliable statements — a pretty strong result, backed up by extensive theory, that we believe will have a substantial impact on XAI since it challenges the concept of “black-box interpretability”. We believe that future work could instead explore constructing networks that incorporate more constraints and assumptions (i.e., networks that are far from being black boxes) and thus enable more reliable predictions.
>
> 9. *With regard to softening the claims (and w.r.t the previous point), one concrete suggestion is, in the conclusion …*
>
> Thanks for the suggestion. We rephrased the conclusion as follows: “Given that our theory proves that visualizing black-box systems via feature visualizations based on activation maximization (i.e., the current dominant paradigm) cannot be guaranteed to be reliable without making strong assumptions about the system, …”.
>
> Thanks again for your helpful comments, questions, and suggestions - we really appreciate it! Kindly let us know whether we have been able to address your comments.

---

> > ### Author Response · Authors · 2023-11-17
> > **Update: new out-of-distribution results included in revision (Appendix E)**
> >
> > *Regarding the sanity check I would be very curious to see the results from other baseline to further explore the causes that could explain the difference.*:
> >
> > We love the suggestion of using out-of-distribution datasets like NINCO to better understand whether feature visualizations fail the sanity check since they might be out-of-distribution. We have now included an analysis on two commonly used and complementary datasets in a dedicated section (Appendix E):
> >
> > 1. **ImageNet-V2**, which has the same classes as original ImageNet and only a small distribution shift as evidenced by a roughly 10-15\% accuracy drop of standard models;
> > 2. **NINCO** as susggested by you, a dedicated out-of-distribution detection dataset that specifically contains no classes that are part of original ImageNet.
> >
> > Appendix E contains results and a brief discussion on those new baselines. Interestingly, we find that the similarity between same-class images of those OOD datasets (including NINCO, which was designed to be a strongly out-of-distribution!) is still a lot higher than the natural-vs-feature-visualization similarity on ImageNet.

---

> > > ### Comment · Reviewer_NDh1 · 2023-11-21
> > > **Response to authors**
> > >
> > > Thank you for the thorough responses.
> > >
> > > This paper has pointed to interesting practical limitations of current Feature visualization (e.g., why they seem to be treated fundamentally differently than natural images) which are deserving of further explorations as this method offers a unique perspective in the current state of deep learning interpretability, despite the other theoretical limitations highlighted in this work.
> > >
> > > Finally, as the authors have addressed most of my concerns, I am raising my score.

---

> > > > ### Author Response · Authors · 2023-11-21
> > > > **Thanks!**
> > > >
> > > > Thanks for your fast response, and we're happy to hear that we have been able to address important concerns! Your detailed review and suggestions have improved our manuscript considerably.

---

### Meta-Review · Area_Chair_X9mu · 2023-12-14

**Metareview:**

The paper proposes a set of empirical experiments and theoretical proofs to show that (1) feature visualization methods are not reliable i.e. it is possible to change a neural network classifier such that even though the new classifier behaves similarly to the original one, its feature visualizations would change arbitrarily; (2) it is not guaranteed for people to be able to predict the classification predictions given the feature visualizations (the impossibility theorem).

All reviewers have positive ratings (6,6,8,8) yet at a fairly low confidence (3/4 reviewers have a confidence score of 3).
Most reviewers agree the need for evaluating and proving the actual utility of feature visualizations is important. Reviewers also agree the paper is well-written. Some reviewers expressed initial concerns about the need for the theory work in this paper (`NDh1` and `MuGq`) but the concerns seem to be ameliorated after the authors' rebuttal.

Although I appreciate the intended mathematical rigor of the work, I find the results are not surprising and weak in terms of impact.
1. First, feature visualizations (Nguyen et al. 2019 or Olah et al. 2017) are themselves the results of non-convex optimization. The resultant images are therefore highly dependent on the hyperparameters including the choice of the image priors (Nguyen et al. 2016; Olah et al. 2017), random seeds, number of optimization steps, the target network, etc.
2. Second, the success or utility of feature visualization demonstrated in the literature so far is entirely empirical and qualitative. Because the target audience of feature visualization is **human**. There needs to be humans in the loop for this evaluation to happen in the first place (and any claims to be scientifically trusted).

Yet, this paper attempts to mathematically prove that it is impossible to guarantee the success of feature visualizations while not taking into account (a) the choice of the image prior (which can entirely change the images) and (b) the human part (which is the factor that entirely determines the success/utility of feature visualizations).

Overall, I vote to `reject` the paper due to the main points above.

**Justification For Why Not Higher Score:**

The paper has a fundamental flaw in the approach and therefore the result is not wrong but not surprising or very useful.

**Justification For Why Not Lower Score:**

N/A

---

### Decision · Program_Chairs · 2024-01-16

Reject